# MANF stimulates autophagy and restores mitochondrial homeostasis to treat autosomal dominant tubulointerstitial kidney disease in mice

Yeawon Kim [1,15], Chuang Li [1,15], Chenjian Gu [1], Yili Fang [1], Eric Tycksen [2], Anuradhika Puri [3], Terri A. Pietka [4], Jothilingam Sivapackiam [5], Kendrah Kidd [6,7], Sun-Ji Park [1], Bryce G. Johnson [8], Stanislav Kmoch [6,7], Jeremy S. Duffield[9], Anthony J. Bleyer [6,7], Meredith E. Jackrel [3], Fumihiko Urano [10], Vijay Sharma [5,11,12], Maria Lindahl [13] & Ying Maggie Chen [1,14] ✉

Misfolded protein aggregates may cause toxic proteinopathy, including autosomal dominant tubulointerstitial kidney disease due to uromodulin mutations (ADTKD-*UMOD*), a leading hereditary kidney disease. There are no targeted therapies. In our generated mouse model recapitulating human ADTKD-*UMOD* carrying a leading *UMOD* mutation, we show that autophagy/mitophagy and mitochondrial biogenesis are impaired, leading to cGAS-STING activation and tubular injury. Moreover, we demonstrate that inducible tubular overexpression of mesencephalic astrocyte-derived neurotrophic factor (MANF), a secreted endoplasmic reticulum protein, after the onset of disease stimulates autophagy/mitophagy, clears mutant UMOD, and promotes mitochondrial biogenesis through p-AMPK enhancement, thus protecting kidney function in our ADTKD mouse model. Conversely, genetic ablation of MANF in the mutant thick ascending limb tubular cells worsens autophagy suppression and kidney fibrosis. Together, we have discovered MANF as a biotherapeutic protein and elucidated previously unknown mechanisms of MANF in the regulation of organelle homeostasis, which may have broad therapeutic applications to treat various proteinopathies.

Misfolded proteins due to genetic mutations and the resultant endoplasmic reticulum (ER) stress represent one important cause of ER storage disease and toxic proteinopathy, including autosomal dominant tubulointerstitial kidney disease due to uromodulin mutations (ADTKD-*UMOD*), Alzheimer's disease[1], amyotrophic lateral sclerosis[2], Wolfram syndrome[3], cystic fibrosis[4] and α1-antitrypsin deficiency[5]. Currently, there is no mechanistic treatment. ADTKD-*UMOD* is a monogenic form of renal tubulointerstitial fibrosis and chronic kidney disease (CKD), which occurs in ~10% of the population associated with significant morbidity and mortality[6]. ADTKD also represents as many as 25% of patients with inherited kidney disease, after exclusion of polycystic kidney disease and Alport syndrome[7]. Uromodulin (Tamm-Horsfall protein) is largely expressed in the thick ascending limb (TAL) tubular epithelial cells. Currently, 135 *UMOD* mutations have been

identified, and some of these mutations have been shown to cause protein misfolding and ER stress, eventually causing TAL damage, inflammatory cell infiltration and fibrosis[8,9]. However, the molecular link between the ER stress activation and renal fibrosis is still missing.

ER protein aggregates can activate the unfolded protein response (UPR). The UPR is regulated by three pathways: inositol-requiring enzyme 1 (IRE1)-spliced XBP1 (XBP1s), protein kinase-like ER kinase (PERK)-activating transcription factor 4 (ATF4), as well as cleavage of 90-kD activating transcription factor 6 (ATF6) to the active 50-kD ATF6. Meanwhile, there is intensive crosstalk between ER stress signaling and the autophagy-lysosomal pathway, and autophagy is a highly conserved protein degradation process responsible for removal of aggregate-prone proteins (aggrephagy) and damaged organelles. The autophagic flux proceeds through several phases: phagophore initiation and nucleation that requires unc-51-like kinase 1 and 2 (ULK1/2)/FIP200/Atg13 and PI3K/Vps34/Vps15/beclin 1, respectively; vesicle elongation forming autophagosomes that requires microtubule-associated proteins 1 A/1B light chain 3B (LC3B) and Atg5-Atg12-Atg16 ubiquitin-conjugation systems; autophagosome maturation; and autophagosome-lysosome fusion forming autolysosomes[10]. The mechanisms that regulate proteostasis of UMOD remain poorly understood and treatment is still lacking.

The selective removal of damaged mitochondria through autophagy is called mitophagy. Mitophagy is mainly mediated by LC3-associated, ubiquitin- and receptor-dependent pathways. PINK1, a mitochondrial Ser/Thr kinase, on the outer membrane of damaged mitochondria can recruit Parkin, an E3 ubiquitin ligase. After ubiquitination, damaged mitochondria are selectively recognized by adapter proteins and engulfed by autophagosomes. Recently, it has been shown that mitofusin 2 (MFN2) is an additional PINK1 substrate for Parkin recruitment besides its role in mitochondrial fusion[11]. Mitophagy receptors include BCL-2/adenovirus E1B 19 kDa protein-interacting protein 3 (BNIP3), BCL-2-interacting protein 3-like (BNIP3L) and FUN14 domain-containing 1 (FUNDC1)[12]. Whether mitophagy is dysregulated in ADTKD-*UMOD* has not been studied.

Mesencephalic astrocyte-derived neurotrophic factor (MANF), an 18 kD, ER soluble protein, can exert protective function in Parkinson's disease and ischemic stroke[13,14], and regulate immune homeostasis in aging and retinal regenerative therapies[15,16] in animal models. It is also upregulated and secreted in response to ER stress. The secreted extracellular MANF can protect cells from stress-induced cell death in experimental models of myocardial infarction[17] and diabetes[18] by inhibiting ER stress-induced apoptosis. We have discovered that secreted MANF can serve as a urinary ER stress biomarker[19] and function as an ER calcium stabilizer for ER-stressed podocytes in vitro[20]. However, the function of MANF has not been investigated in kidney disease in vivo.

By using CRISPR/Cas9, we generated an ADTKD-*UMOD* mouse model carrying UMOD p.Tyr178-Arg186 del, the mouse equivalent of the human UMOD p.His177-Arg185 del in-frame deletion, one of the most prevalent mutations in human ADTKD-*UMOD*[21]. To assess the functional role of MANF in ADTKD, we generated both TAL cell-specific MANF knockout and inducible tubular cell-specific MANF transgenic mice. We demonstrate that MANF expression is increased exclusively in ER-stressed TAL epithelium. MANF deletion in mutant TALs exacerbates autophagy failure and renal fibrosis. Conversely, MANF overexpression in renal tubules augments autophagy/mitophagy, improves mitochondrial function, and inhibits tubular cGAS-STING dependent inflammation, thus attenuating kidney injury and fibrosis. Taken together, our findings reveal the role of MANF in maintaining functional autophagy, and highlight the important therapeutic function of MANF by regulating organelle homeostasis for the treatment of ADTKD and possibly other toxic proteinopathy.

## Results

### Generation of UMOD Y178-R186 deletion mice to recapitulate human ADTKD-*UMOD*

We used CRISPR/Cas9 with one guide RNA (sgRNA) targeting the *Umod* gene's 5′-TATGAGACCCTGACTGAGTACTGGCGC-3′ to delete amino acids YETLTEYWR in exon 3 (Fig. 1a). The sgRNA and Cas9 protein that were complexed to generate the ribonucleoprotein, along with a 200-bp single-stranded donor DNA containing the Y178-R186 del, 27-bp deletion of *Umod*, were injected into C57BL/6 J mouse embryos at the single-cell stage. Founder genotyping was performed by deep sequencing, and 32 founders were identified. The positive founders were further crossed to C57BL/6 J mice to generate heterozygous F1 offspring, which were also deep sequenced to confirm correctly targeted alleles. The heterozygous mice of the F1 generation were found to have a germline transmission of 50%.

A cohort of heterozygous (*Umod* [DEL/+]) and WT littermates were followed until 24 weeks of age. Blood urea nitrogen (BUN) was slightly increased at 3 weeks, significantly elevated by 4 weeks, and progressively increased between 4 and 24 weeks in the mutants (Fig. 1b). As expected, co-immunofluorescence (IF) staining of UMOD and the ER stress marker BiP in kidney sections at 16 weeks highlighted discontinuous and punctate UMOD protein aggregates in the ER in the mutant mice (Fig. 1c, insets), consistent with our previous finding from a human kidney biopsy carrying the *UMOD* H177-R185 del[22]. In addition, the mutant UMOD induced ER stress (Fig. 1c). Meanwhile, we also noted that the mutant UMOD aggregation occurred by 3 weeks (Supplementary Fig. 1a). In agreement with the staining result, WB revealed elevation of the under-glycosylated form of mutant UMOD due to ER retention, which migrated faster than the WT UMOD band, as early as 3 weeks and much more prominent ER accumulation at 16 weeks (Fig. 1d). In contrast, urinary UMOD excretion was markedly decreased in the mutants, along with increased excretion of our recently identified urinary ER stress biomarkers, secreted BiP[23] and cysteine-rich with EGF-like domains 2 (CRELD2)[22] as early as 3 weeks (Fig. 1e). Masson's trichrome staining showed progressive tubulointerstitial fibrosis during the disease progression, with the fibrosis accentuated in the corticomedullary junction (Fig. 1f). Consistent with this observation, whole-kidney mRNA and protein quantifications and IF staining demonstrated a progressive increase in fibrosis markers in DEL/+ mice. By 12 weeks, fibronectin (FN) expression was increased (Supplementary Fig. 1b). At 16 weeks, a marked increase in both protein levels of FN and laminin (LN) (Fig. 1g, h, j) and transcript levels of smooth muscle actin (SMA) (*Acta2*) and collagen I α1 (*Col1a1*) (Fig. 1i) were observed. At 24 weeks, immunoblots of LN, SMA and FN (Supplementary Fig. 1c), q-PCR of *Col1a1* and *Tgfb* (Supplementary Fig. 1d) and IF of SMA (Fig. 1k) analyses continued to display the substantial increase in fibrosis in DEL/+ mice. Overall, we have successfully established an ADTKD-*UMOD* mouse model harboring a pathogenic human mutation that recapitulates monogenic CKD in the patients.

### Impaired autophagy in TALs expressing the mutant UMOD in ADTKD-*UMOD* mice

To investigate the molecular mechanisms underlying the disease pathogenesis, first, we determined which UPR branch was activated. WB demonstrated that the ATF6 pathway was activated by 12 weeks (Fig. 2a, arrow), and persisted through 24 weeks in the mutant kidneys, whereas the other two UPR branches were not involved (Supplementary Fig. 2a–c). Next, we examined autophagic activity in the mutant kidneys. As elevated LC3B-II levels, which indicate autophagosomal accumulation, can signify the changes in both formation and degradation of the autophagosomes, p62, the selective degradation substrate of autophagy that links ubiquitinated proteins to LC3, also needs to be assessed. Accumulation of p62 protein in the absence of increased transcription indicates that autophagy clearance is inhibited.

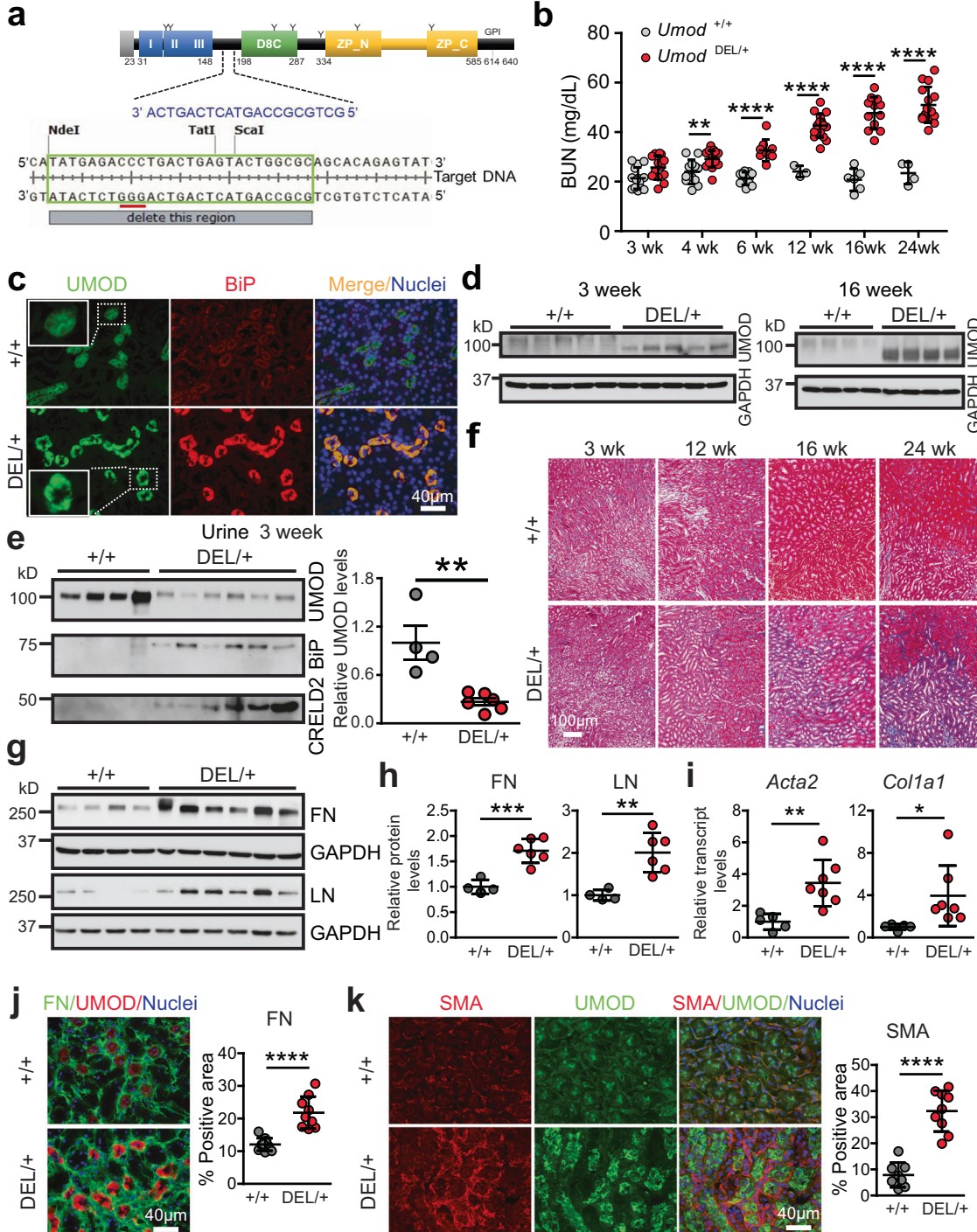

**Fig. 1 | Generation of a mouse model that recapitulates human ADTKD-*UMOD*.**
**a** The CRISPR target sites are outlined in green, the sgRNA sequences are marked in blue, and protospacer adjacent motifs (PAMs) are underlined in red. **b** BUN measurements over a 24-week period (at 3, 4, 6, 12, 16, 24 weeks, $n = 14$, 12, 11, 3, 6, 4 for WT, $n = 13$, 17, 9, 17, 12, 15 for DEL/+). Two-tailed $t$-test, Mean ± SD. **$p = 0.0019$ (4 weeks), ****$p < 0.0001$ (for 6–24 weeks). **c** IF images of kidney sections stained for UMOD and BiP with nuclei counterstain (blue) at 16 weeks. **d** Immunoblot to detect UMOD protein from whole-kidney tissues. **e** WBs of urine UMOD, BiP and CRELD2, indexed to urine creatinine (4 μg), from WT ($n = 4$) and DEL/+ ($n = 6$) mice at age 3 weeks, with densitometry analysis of urine UMOD. Two-tailed $t$-test, Mean ± SD. **$p = 0.003$. **f** Trichrome staining of WT and DEL/+ kidney sections from 3–24 weeks. **g**, **h** WBs of whole-kidney lysates from WT ($n = 4$) and DEL/+ ($n = 6$)

mice at 16 weeks to detect FN and LN (**g**) with densitometry analysis (**h**). Two-tailed $t$-test, Mean ± SD. ***$p = 0.0007$ (FN), **$p = 0.0033$ (LN). **i** Q-PCR of transcript levels of *Acta2* and *Col1a1*, normalized to 18 s, in WT ($n = 5$) and DEL/+ ($n = 7$) kidneys at age 16 weeks. Two-tailed $t$-test, Mean ± SD. **$p = 0.0053$ (*Acta2*), *$p = 0.0465$ (*Col1a1*). **j** Dual IF staining of FN (green) and UMOD (red) with nuclei counterstain (blue) on kidney sections from WT and DEL/+ mice at 16 weeks with quantification. $n = 10$, 9 images for WT, DEL/+ kidneys, respectively. Two-tailed $t$-test, Mean ± SD. ****$p < 0.0001$. **k** Dual IF staining of SMA and UMOD with nuclear counterstain (blue) on kidney sections from WT and DEL/+ mice at 24 weeks with quantification. $n = 8$, 9 images for WT, DEL/+ kidneys, respectively. Two-tailed $t$-test, Mean ± SD. ****$p < 0.0001$. Source data are provided as a Source Data file.

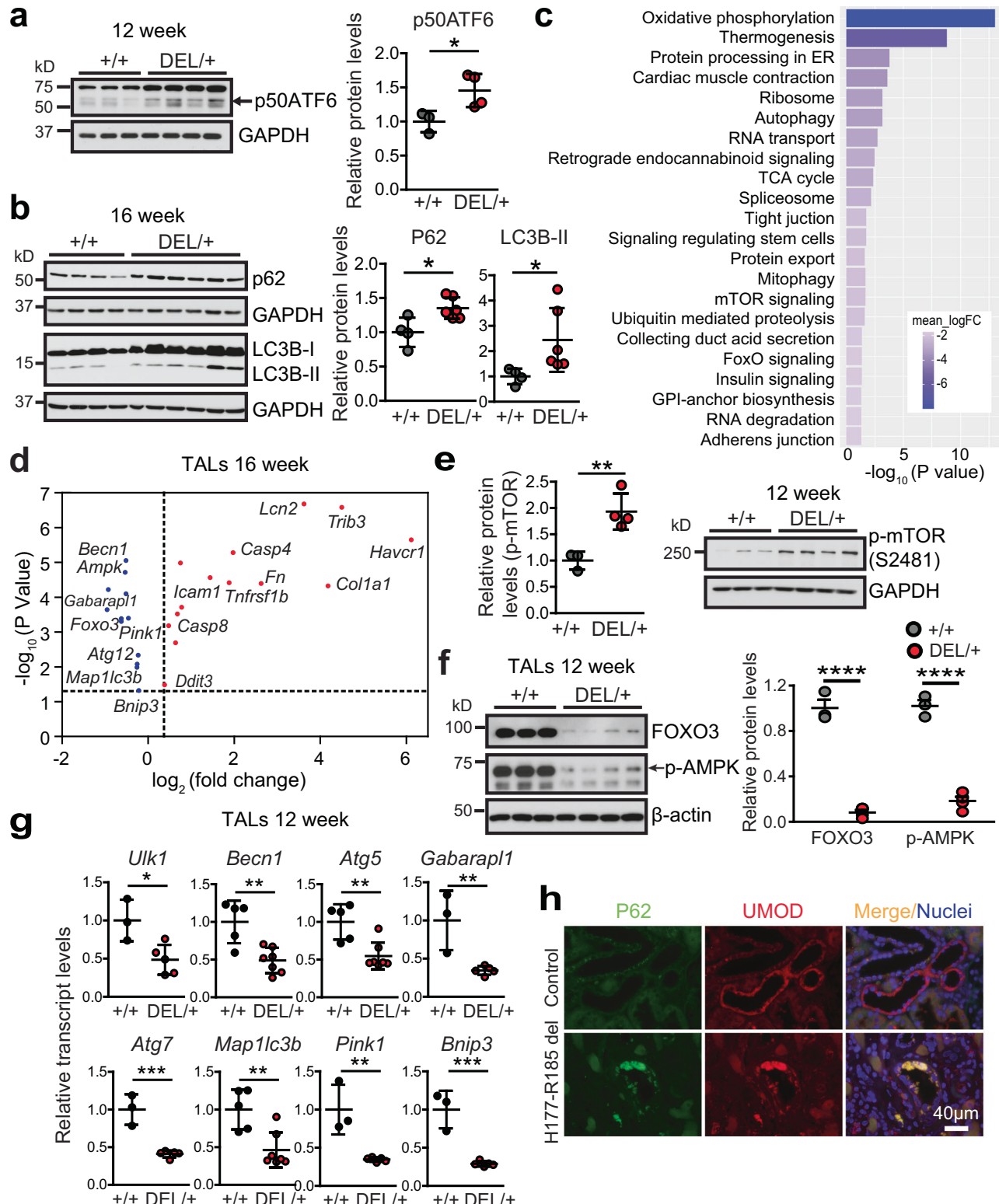

During the disease progression, protein abundance of p62 was increased by 12 weeks (Supplementary Fig. 2d), and p62 and LC3B-II (LC3B-I inactive, LC3B-II active) persisted through 16–24 weeks (Fig. 2b and Supplementary Fig. 2e) without transcriptional upregulation of p62 (Supplementary Fig. 2f) in the mutant kidneys, demonstrating impaired autophagic degradation of the misfolded protein.

To gain further insight into the mechanisms underlying the defective autophagic activity and its functional impact on the pathogenesis of disease, we performed RNA sequencing (RNA-seq) of mRNA isolated from UMOD-producing TAL cells purified from $Umod^{DEL/+}$ and $Umod^{+/+}$ littermates at 16 weeks, the full-blown stage of the disease. Principal components analysis of RNA-seq expression patterns across these two groups revealed clustering within each genotype, suggesting two very distinct cell populations (Supplementary Fig. 3a). In addition, our analysis revealed that 9566 genes were differentially expressed between mutant and WT TAL cells (Benjamini-Hochberg

**Fig. 2 | Impaired autophagy in the mutant TALs in ADTKD-*UMOD*. a** Whole-kidney lysates from *Umod* [+/+] (*n* = 3) and *Umod* [DEL/+] (*n* = 4) mice at 12 weeks were analyzed by WB for p50ATF6 (arrow) with densitometry analysis. Two-tailed *t*-test, Mean ± SD. *\**p* = 0.0376. **b** WBs of whole-kidney lysates (WT, *n* = 4; *Umod* [DEL/+], *n* = 6) at 16 weeks to detect the autophagy mediators p62 and LC3B with densitometry analysis. Two-tailed *t*-test, Mean ± SD. *\**p* = 0.016 (for p62), 0.049 (LC3B-II). **c**, **d** RNA-seq analysis of mutant and WT TALs at 16 weeks. *n* = 5/group. Benjamini-Hochberg adjusted *p* < 0.05. **c** KEGG pathway analysis of the log2 fold-changes in TAL cells showed the top 22 downregulated pathways. GAGE analysis. **d** RNA-seq analysis of genes for autophagy, ER stress response, inflammation, apoptosis, kidney injury and fibrosis in the mutant TALs compared with WT TALs. Limma analysis for differential gene expression. **e** WB from *Umod* [+/+] (*n* = 3) and *Umod* [DEL/+] (*n* = 4) kidneys for levels of p-mTOR (S2481) at 12 weeks with densitometry analysis.

Two-tailed *t*-test, Mean ± SD. *\*\**p* = 0.0081. **f** WBs from *Umod* [+/+] (*n* = 3) and *Umod* [DEL/+] (*n* = 4) TALs for levels of FOXO3 and p-AMPK (arrow) at 12 weeks with densitometry analysis. Two-tailed *t*-test, Mean ± SD. *\*\*\*\**p* < 0.0001. **g** Transcript analysis of a panel of autophagy and mitophagy-related genes, normalized to 18 s, from TALs at 12 weeks. WT (*n* = 3) and *Umod* [DEL/+] (*n* = 5) for *Ulk1*, *Gabarapl1*, *Atg7*, *Pink1* and *Bnip3*; WT (*n* = 5) and *Umod* [DEL/+] (*n* = 7) for *Becn1*, *Atg5* and *Map1lc3b*. Two-tailed *t*-test, Mean ± SD. *p* = 0.02 (for *Ulk1*), 0.0027 (*Becn1*), 0.0034 (*Atg5*), 0.0079 (*Gabarapl1*), 0.0006 (*Atg7*), 0.0039 (*Map1lc3b*), 0.0032 (*Pink1*), 0.0005 (*Bnip3*). **h** Representative IF images of human kidney biopsies obtained from a patient with p.H177-R185 del and from a normal kidney, stained for p62 (green) and UMOD (red) with nuclei counterstain (blue) on paraffin kidney sections. Scale bar, 40 μm. Source data are provided as a Source Data file.

adjusted *p* ≤ 0.05) (Supplementary Fig. 3b). As depicted in the volcano plot, among these 9566 differentially expressed transcripts, including 4706 upregulated and 4860 downregulated, 244 genes were up-regulated and 149 were down-regulated by 4-fold or more in the mutant UMOD-expressing TAL cells (Supplementary Fig. 3b). Kyoto Encyclopedia of Genes and Genomes (KEGG) pathway analysis for mutant versus WT TAL cells revealed top 22 downregulated pathways, which are related to mitochondria oxidative phosphorylation, ATP generation (thermogenesis), ER proteostasis, autophagy, TCA cycle, mitophagy, mTOR and FOXO signaling, as well as ubiquitin-mediated protein degradation (Fig. 2c). Additional RNA-seq analysis of dysregulated genes specifically in mutant TALs at the 16-week time point confirmed a significant suppression of autophagy-related genes, including *Becn1*, *Ampk*, *Gabarapl1* encoding ATG8, *Foxo3*, *Pink1*, *Atg12*, *Map1lc3b* encoding LC3B, and *Bnip3* (Fig. 2d and the source data are provided as a Source Data file). We also observed a marked activation of genes involved in epithelial injury (*Lcn2* encoding neutrophil gelatinase-associated lipocalin (NGAL), *Havcr1* encoding Kim1), matrix dysregulation and fibrosis (*Col1a1*, *Fn*), inflammation (*Tnfrsf1b*, *Icam1*), ER stress-induced apoptosis (*Trib3*, *Ddit3*), as well as *Casp4* and *Casp8* (Fig. 2d and the source data are provided as a Source Data file). Q-PCR analysis of the isolated TAL cells at 16 weeks confirmed repression of autophagy in the mutant TALs (Supplementary Fig. 4).

mTOR negatively regulates autophagy through phosphorylation at Ser757 to inhibit ULK1 activity and autophagosome formation. Consistent with the RNA-seq data (Fig. 2c), WB showed increased levels of active p-mTOR (S2481) in the mutant kidneys at 12 weeks, the early stage of the disease (Fig. 2e). To further understand the molecular control of the dysfunctional autophagy in the mutant TALs, we purified WT and mutant TAL cells at early stage of the disease, 12 weeks. Given that in the mutant TALs RNA-seq revealed inhibited FOXO signaling (Fig. 2c), a family of critical transcriptional factors that induce autophagy by transactivating expression of genes encoding induction, nucleation, elongation and fusion of the autophagic process[24], we examined expression of FOXO3a. The mutant TAL cells exhibited remarkably decreased FOXO3 expression at 12 weeks (Fig. 2f). Moreover, protein expression of the active, phosphorylated AMP-activated protein kinase (p-AMPK) at Thr-172, which can phosphorylate FOXO3 at Ser413 or Ser588, promoting nuclear accumulation and stabilization and preventing cytoplasmic degradation of FOXO3[24], as well as block mTOR activity, was also dramatically reduced in the mutant TAL cells at 12 weeks (Fig. 2f, arrow). Besides the upstream regulator, a further transcriptional analysis of FOXO3 downstream targets in the mutant UMOD epithelium at 12 weeks revealed inhibition of a panel of autophagy machinery genes related to initiation (*Ulk1*), nucleation (*Becn1*) and elongation (*Atg5*, *Gabarapl1*, *Atg7* and *Map1lc3b*) of autophagosomes, as well as suppression of *Pink1* and *Bnip3*, which mediates mitophagy[25] (Fig. 2g). Finally, we confirmed our results obtained from the mouse model in human kidney biopsies. Co-IF staining of p62 and UMOD clearly showed increased p62 aggregates in the UMOD-positive epithelium in a patient harboring H177-R185 del

compared with a healthy control (Fig. 2h). Taken together, these findings demonstrate that autophagy is actively suppressed in the UMOD proteinopathy model.

## Defective mitophagy and mitochondrial biogenesis in the mutant TALs

Mitophagy depends on the activity of the autophagy machine, and p-AMPK/FOXO3 regulate mitophagy as well[26], we therefore set out to study whether mitophagy is inhibited in the mutant TALs. Heatmap of the 16-week RNA-seq data showed that expression of multiple genes involved in mitophagy was significantly downregulated in the mutant vs. WT TALs (Benjamini-Hochberg adjusted *p* ≤ 0.05) (Fig. 3a). Immunoblot analysis of isolated TALs at 12 weeks further validated that both ubiquitin-mediated PINK1/MFN2/Parkin and receptor-mediated BNIP3/BNIP3L/FUNDC1 pathways were markedly inhibited in the mutant TALs (Fig. 3b and Supplementary Fig. 5a). Moreover, Mitochondrial fission is required for mitophagy and allows damaged mitochondria to be eliminated by mitophagy[27]. In agreement with suppressed mitophagy, we also observed decreased expression of mitochondrial fission protein 1 (FIS1), a mitochondrial fission marker, in the mutant TALs at the early stage of disease, as evidenced by the heatmap (Fig. 3a) and WB (Fig. 3c and Supplementary Fig. 5b).

Given that during stress, AMPK also activates peroxisome proliferator-activated receptor-γ (PPARγ) co-activator 1α (PGC1α), which regulates mitochondrial biogenesis genes through interaction with PPARγ or estrogen-related receptors (ERRs)[26], we next examined mitochondrial biogenesis in our ADTKD-*UMOD* mouse model. At 12 weeks, both PGC1α and its downstream target mitochondrial transcription factor A (TFAM), a key regulator of mitochondrial gene expression[28], were substantially repressed in the mutant TALs compared with WT TALs (Fig. 3d and Supplementary Fig. 5c). Consequently, transcript levels of mitochondrial DNA (mtDNA), such as *mt-Co1*, *mt-Rnr2*, *mt-Nd4* and *mt-Cytb* in the mutant kidneys at 16 weeks (Fig. 3e), as well as protein levels of mitochondrial oxidative phosphorylation (OXPHOS)-related proteins, by utilizing a total OXPHOS WB antibody cocktail, in the mutant TALs at 12 weeks (Fig. 3f and Supplementary Fig. 5d) were significantly lower than controls. In contrast, in UMOD-negative cells, no difference in expression of the subunits of electron transport complex (ETC) proteins was noted between WT and mutant kidneys (Fig. 3f). In line with the WB result, the KEGG analysis clearly depicted downregulation of all five ETCs in the mutant TALs as compared to WT TALs at 16 weeks (Supplementary Fig. 6).

To determine the activity of ETCs and mitochondrial respiratory function, we performed high-resolution respirometry using an ORO-BOROS Instruments Oxygraph-O2k system in freshly isolated WT and mutant TAL cells at 16 weeks. To measure $O_2$ flux, different substrates were added sequentially, including glutamate, malate and pyruvate (G + M + P), adenosine diphosphate (ADP) and succinate, followed by the OXPHOS uncoupler carbonyl cyanide-p-trifluoromethoxyphenylhydrazone (FCCP) to determine leak respiration, complex I activity, complex I + II activity and maximal ETC

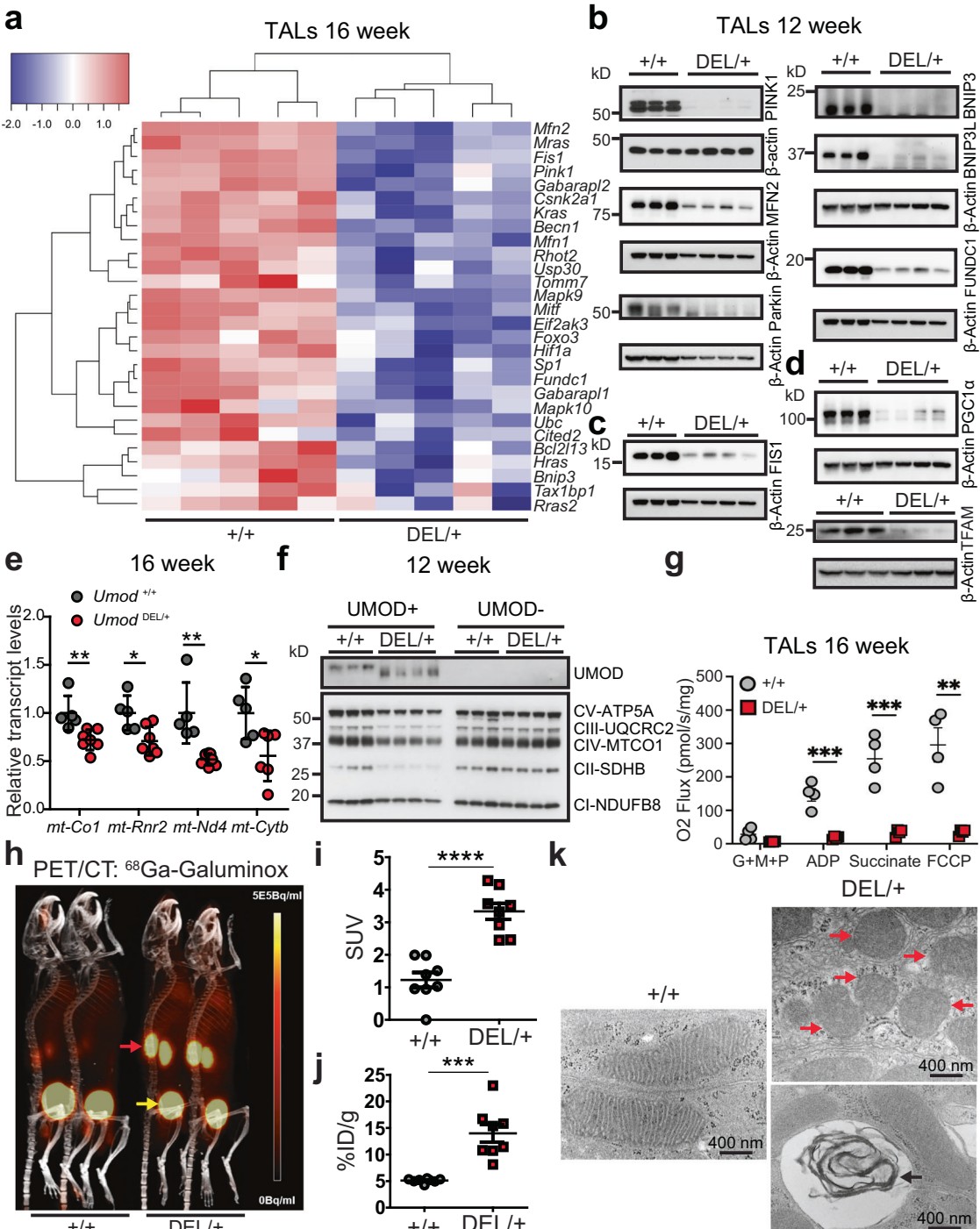

**Fig. 3 | Deficient mitophagy and dysfunctional mitochondrial biogenesis in the mutant TALs in ADTKD.** a Heatmap of downregulated mitophagy-associated genes with Benjamini-Hochberg adjusted $p \leq 0.05$. GAGE analysis. RNA was isolated from *Umod* [+/+] and *Umod* [DEL/+] TALs from 16-week-old mice. $n = 5$ mice/group. **b** Representative immunoblot analysis monitoring mitophagy pathways in *Umod* [+/+] and *Umod* [DEL/+] TALs at 12 weeks. $n = 3-4$ mice/genotype. **c** Representative immunoblot analysis monitoring FIS1 in *Umod* [+/+] and *Umod* [DEL/+] TALs at 12 weeks. $n = 3-4$ mice/genotype. **d** Representative immunoblot analysis monitoring mitochondrial biogenesis markers in *Umod* [+/+] and *Umod* [DEL/+] TALs at 12 weeks. $n = 3-4$ mice/genotype. **e** Relative mRNA levels of mtDNA genes, normalized to 18 s, from *Umod* [+/+] ($n = 5$) and *Umod* [DEL/+] ($n = 6-7$) kidneys at 16 weeks. Two-tailed *t*-test, Mean ± SD. $p = 0.0067$ (for *mt-Co1*), 0.0119 (*mt-Rnr2*), 0.0026 (*mt-Nd4*), 0.0219 (*mt-Cytb*). **f** Representative immunoblot analysis monitoring ETC protein expression in UMOD-positive and UMOD-negative tubular epithelium from *Umod* [+/+] and *Umod* [DEL/+] kidneys at 12 weeks. $n = 3-4$ mice/genotype. **g** Measurement of mitochondrial

respiration using an OROBOROS Oxygraph system in permeabilized WT and *Umod* [DEL/+] TALs at 16 weeks following sequential additions of G + M + P; ADP; succinate and FCCP. $n = 4$/genotype. Two-tailed *t*-test, Mean ± SD. $p = 0.0005$ (for ADP), 0.0008 (Succinate), 0.0023 (FCCP). **h** PET/CT images post-[68]Ga-Galuminox injection were acquired from *Umod* [+/+] and *Umod* [DEL/+] mice at 16 weeks. Uptake of radiotracers in the kidneys (red arrows) and retention in the bladders (yellow arrows) were shown. **i** SUV analysis of [68]Ga-Galuminox uptake in kidneys of the indicated genotypes. $n = 8$ kidneys/group. Two-tailed *t*-test, Mean ± SD. ****$p < 0.0001$. **j** Post-PET imaging biodistribution data (%ID/g) of *Umod* [+/+] and *Umod* [DEL/+] kidneys at 16 weeks. $n = 8$ kidneys/group. Two-tailed *t*-test, Mean ± SD. ***$p = 0.00011$. **k** TEM ultrastructural analysis on the renal tubules from *Umod* [+/+] and *Umod* [DEL/+] mice at 16 weeks. Red arrows indicate aggregates of swollen mitochondria with disrupted cristae, and black arrow indicates "myelin body". Scale bar, 400 nm. Source data are provided as a Source Data file.

capacity, respectively. As shown in Fig. 3g, oxygen consumption after addition of ADP and succinate substrates, as well as FCCP was significantly decreased in the mutant TALs compared to that in WT TALs, indicating disruption of mitochondrial respiratory function in the mutant TALs (Fig. 3g). By contrast, there was no difference in oxygen consumption for UMOD-negative cells between WT and mutant kidneys at 16 weeks (Supplementary Fig. 7a).

Mitophagy blockade can lead to accumulation of damaged, reactive oxygen species (ROS)-generating mitochondria. Meanwhile, lack of efficient oxidative phosphorylation due to impaired mitochondrial biogenesis can result in overproduction of mitochondrial ROS. Thus, we anticipated increased accumulation of mitochondrial ROS in the mutant kidneys. To prove our hypothesis unambiguously, we employed noninvasive, sensitive, and quantitative PET/CT molecular imaging to detect mitochondrial ROS by utilizing our recently developed mitochondrial ROS radiotracer $^{68}$Ga-Galuminox[29,30]. This is a $^{68}$Ga-radiotracer (incorporated with a nonconventional, generator-produced isotope) capable of detecting mitochondrial ROS in live animals. Representative preclinical PET/CT images (summation of frames over 30–45 min) post-administration of $^{68}$Ga-Galuminox were shown in Fig. 3h (red arrow). $^{68}$Ga-Galuminox showed a 2.73 fold higher uptake in kidneys of $Umod^{DEL/+}$ mice (standard uptake value (SUV): $3.33 \pm 0.25$, $n = 8$) compared with WT littermates (SUV: $1.22 \pm 0.23$, $n = 8$, $p < 0.0001$) (Fig. 3i). For further correlating preclinical PET imaging data, post-imaging biodistribution studies were also conducted and percentage of activity remained in kidneys at 1 h post injection of the radiotracer was quantified as the percentage injected dose (% ID) per gram of kidney. $^{68}$Ga-Galuminox demonstrated that compared with the WT kidneys (% ID/g: $5.1 \pm 0.14$, $n = 8$), the radiotracer was retained 2.7-fold higher in the mutant kidneys (% ID/g: $13.99 \pm 1.66$, $n = 8$, $p < 0.001$) (Fig. 3j). We further confirmed the PET/CT findings by staining the frozen kidney sections of WT and DEL/+ mice at 16 weeks of age with cell permeant chloromethyl-2′,7′-dichlorodihydro-fluorescein diacetate (CM-$H_2$DCFDA), a chemically reduced form of fluorescein used as an indicator for ROS (Supplementary Fig. 7b).

Consistent with these biochemical, functional and in vivo molecular imaging studies, transmission electron microscopy (TEM) ultrastructural analysis of mitochondrial integrity showed that mitochondria were aligned in well-preserved rows in normal renal tubules (Fig. 3k). Mutant kidneys at 16 weeks showed disorganized mitochondrial arrays and aggregates of swollen mitochondria with disrupted cristae (Fig. 3k, red arrows), as well as "myelin body" suggesting possible lysosome damage (Fig. 3k, black arrow). Collectively, these data indicate impaired mitophagy and mitochondrial biogenesis/OXPHOS, leading to increased level of mitochondrial ROS and aberrant mitochondrial ultrastructure in the mutant TALs.

### Increased inflammation and activation of cGAS-STING signaling in ADTKD-*UMOD* mice

Gene Ontology (GO) analysis of the log2 fold-changes for mutant versus WT TAL cells at 16 weeks demonstrated that the most upregulated pathways in the mutant TAL cells were related to immune response and extracellular matrix organization (Fig. 4a). Q-PCR analysis of the isolated TAL cells at 16 weeks confirmed activation of inflammatory genes, such as tumor necrosis factor (TNFα), interleukin-1β (IL-1β), interleukin-6 (IL-6), intercellular adhesion molecule 1 (ICAM1) and chemokine (C-C motif) ligand 2 (CCL2) in the mutant TALs (Fig. 4b). Consistent with increased transcript levels of inflammatory cytokines, hematoxylin & eosin (H&E) staining (Fig. 4c, red arrow) at 24 weeks highlighted kidney ingress of massive inflammatory cells and few kidney cyst (Fig. 4c, black arrow), reminiscent of findings noted in human kidney biopsies[22]. Moreover, co-IF staining of the macrophage marker F4/80 with UMOD showed macrophage infiltration surrounding mutant TAL tubules at 24 weeks (Fig. 4d, e).

To explore the molecular mechanism underpinning the increased immune response in the mutant kidneys, we examined cGAS (cyclic guanosine monophosphate-adenosine monophosphate synthase)-stimulator of interferon genes (STING) signaling that may be activated following the failure of mitochondrial quality control (Fig. 3). Cytosolic mtDNA can be sensed by cGAS, which induces synthesis of cyclic guanosine adenosine monophosphate (cGAMP). cGAMP binds to ER adapter protein STING, and activated STING migrates from ER to the Golgi apparatus. During this process, STING can recruit and activate TANK-binding kinase 1 (TBK1), which in turn, activates the downstream NF-κB and interferon (IFN) regulatory factor 3 (IRF3) signal cascades, thus inducing expression of inflammatory factors and type I IFN to strengthen immune responses[31]. Indeed, STING, pTBK1/TBK1 ratio and their downstream targets p-NF-κB p65 and p-IRF3 exhibited increased abundance in the mutant TALs compared with WT TALs at 16 weeks (Fig. 4f). It has been shown that disruption of mitochondrial integrity and activation of STING and inflammation in renal tubules strongly contribute to cell death and kidney failure[32,33]. Similarly, we observed a prominent cleavage of procaspase-9 in the mutant kidneys at 16 weeks, indicating activation of the mitochondria-dependent apoptotic pathway (Fig. 4g). Ultimately, the executioner caspase-3 was activated at 16 weeks (Fig. 4g), and TUNEL staining easily detected increased apoptosis in both UMOD⁺ (Fig. 4h, white arrows) and UMOD⁻ cells (Fig. 4h, red arrows) at 24 weeks. In summary, a defective engagement of mitophagy and mitochondrial biogenesis in response to ER stress triggered by the misfolded UMOD stimulates cGAS-STING signaling and innate immune response, which may eventually lead to tubular cell death and fibrosis in ADTKD-*UMOD*.

### mtDNA leakage into the cytosol activates cGAS-STING in ADTKD-*UMOD*

To directly demonstrate that STING pathway is activated by cytosolic leakage of mtDNA and recapitulate the in vivo findings in our mouse model, we generated a cellular model of ADTKD-*UMOD* in HEK 293 cells by transducing the cells with lentivirus expressing WT or mutant UMOD (H177-R185 del) fused to N-terminal GFP. It has been shown previously that primary TAL cells lose UMOD expression after 10 days of in-vitro culturing[9], and that immortalized mouse TAL cell line does not express UMOD[34]. Thus, *UMOD* expression was placed under the control of the cytomegalovirus (CMV) promoter and secretion of the GFP-UMOD fusion proteins was directed by the endogenous secretory signal peptide of UMOD. Confocal live imaging of stable cell lines showed clear membrane enrichment for WT UMOD and diffuse reticular distribution in the cytoplasm for the mutant UMOD, whereas GFP alone was located in both cytoplasm and nucleus (Fig. 5a). Q-PCR demonstrated that the mRNA expression levels of *UMOD* in the two cell lines were comparable (Fig. 5b). Immunoblot analysis showed elevation of mutant form of GFP-UMOD by using both anti-UMOD and anti-GFP antibodies in the mutant H177-R185 del UMOD cell line (DEL) compared to WT cell line, associated with reduced secretion of the GFP-tagged mutant UMOD into the cell culture medium (Fig. 5c). IF staining following fixation and permeabilization of cell membrane of the stably transduced HEK 293 cells confirmed ER retention of the mutant UMOD by its increased co-localization with the ER marker calnexin as compared to WT UMOD (Fig. 5d).

DEL cells, as compared to WT cells, exhibited increased expression of BiP and selective activation of the p50ATF6 branch without involvement of XBP1s and ATF4 branches (Fig. 5e and Supplementary Fig. 8a). DEL cells also showed increased protein levels of p62, LC3B-II and p-mTOR (S2481) (Fig. 5f and Supplementary Fig. 8b) without transcriptional upregulation of p62 (Fig. 5g), as well as decreased expression of PINK1/Parkin/FUNDC1 (Fig. 5h and Supplementary Fig. 8c) compared to WT cells, indicating insufficient autophagy/mitophagy. Meanwhile, decreased protein levels of PGC1α and TFAM,

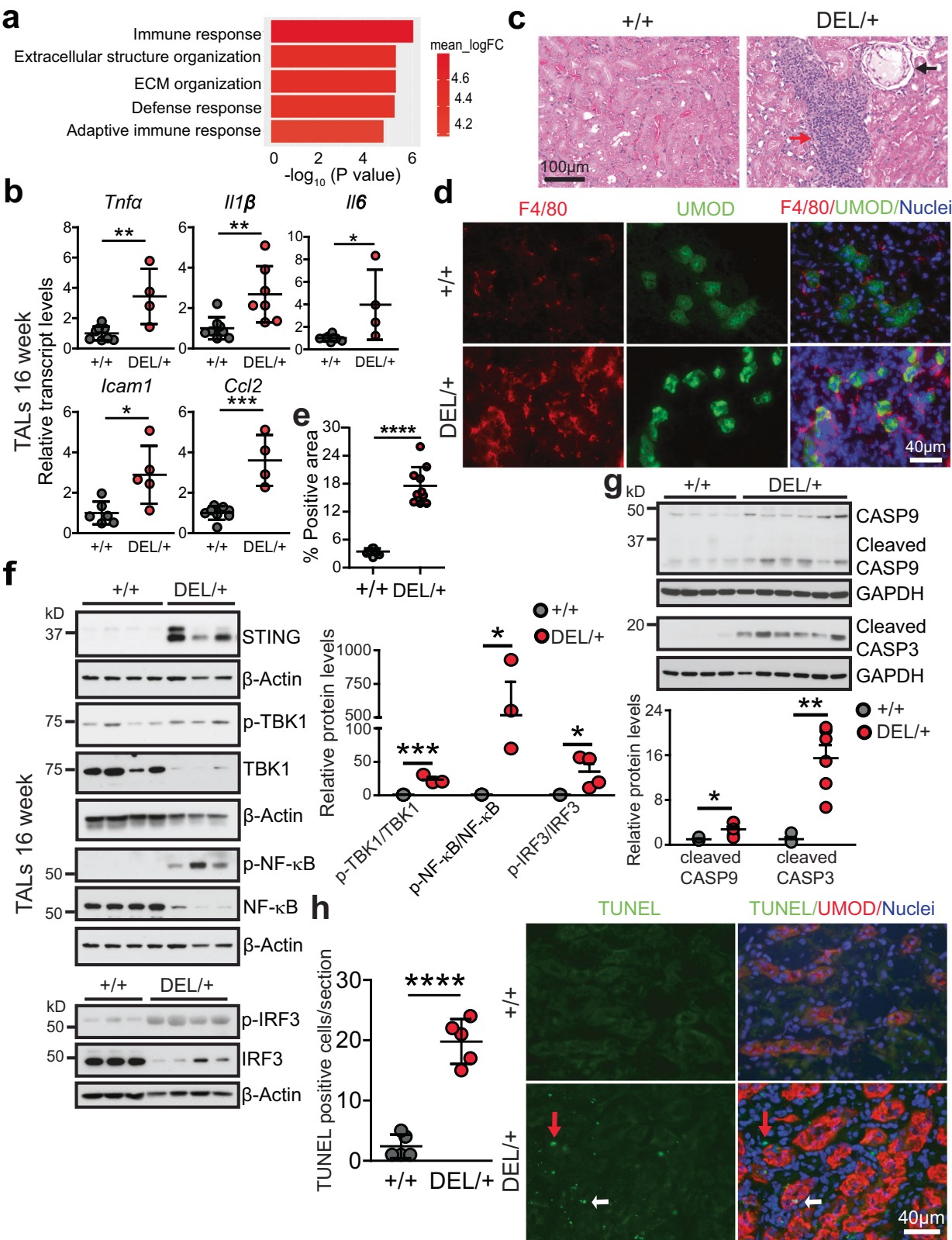

which reflected dysfunctional mitochondrial biogenesis, were noted in DEL cells vs. WT cells (Fig. 5i and Supplementary Fig. 8d). Together, we conclude that the stable cell model simulates the findings in our ADTKD-*UMOD* mouse model well.

By utilizing the cell model, we isolated cytosolic fractions and directly measured release of mtDNA into the cytosol by normalizing the copy number of mtDNA encoding cytochrome *c* oxidase I to

nuclear DNA encoding 18 S ribosomal RNA. As shown in Fig. 5j, the mtDNA leakage into the cytosol was significantly higher in DEL cells relative to that in the WT cells. Furthermore, consistent with the mouse model, activated STING/p-TBK1/p-NF-κB/p-IRF3 signaling (Fig. 5k and Supplementary Fig. 8e) and increased transcript levels of inflammatory cytokines, including *TNFα*, *IL6*, *IL1β*, *CCL2* and *ICAM1* (Fig. 5l) were observed in the mutant cells compared with WT cells. We next

**Fig. 4 | Activation of STING signaling, increased inflammation and apoptosis in ADTKD-*UMOD*. a** RNA-seq of WT and mutant TALs at 16 weeks. *n* = 5/group. GO biological process perturbation bar plot showed the most upregulated pathways in the mutant vs. WT TALs at 16 weeks. GAGE analysis. **b** Quantitative PCR for proinflammatory cytokines, normalized to β-actin, in isolated TALs (WT: *n* = 6 for *Icam1* and *n* = 8 for other genes; *Umod* DEL/+: *n* = 4 for *Tnfa*, *Il6* and *Ccl2*, *n* = 7 for *Il1b* and *n* = 5 for *Icam1*) at 16 weeks. Two-tailed *t*-test, Mean ± SD. *p* = 0.0039 (for *Tnfa*), 0.0075 (*Il1b*), 0.0179 (*Il6*), 0.0151 (*Icam1*), 0.0002 (*Ccl2*). **c** H&E staining of paraffin kidney sections from *Umod* +/+ and *Umod* DEL/+ mice at 24 weeks. Red arrow indicates interstitial inflammatory infiltration. Black arrow indicates a cortical renal cyst. **d**, **e** Representative IF staining of macrophage (F4/80) and UMOD with nuclei counterstain (blue) on frozen kidney sections from *Umod* +/+ and *Umod* DEL/+ mice at 24 weeks (**d**) with quantification (**e**). *n* = 10 images/genotype. Two-tailed *t*-test,

Mean ± SD. ****p* < 0.0001. **f** Immunoblot analysis monitoring STING signaling in *Umod* +/+ and *Umod* DEL/+ TALs at 16 weeks, including STING, p-TBK1/TBK1, p-NF-κB p65/NF-κB p65 (WT: *n* = 4; *Umod* DEL/+: *n* = 3) and p-IRF3/IRF3 (WT: *n* = 3; *Umod* DEL/+: *n* = 4) with densitometry analysis. Two-tailed *t*-test, Mean ± SD. *p* = 0.0008 (for p-TBK1/TBK1), 0.0493 (p-NF-κB/NF-κB), 0.0484 (p-IRF3/IRF3). **g** Immunoblot analysis monitoring cleaved CASP9 and CASP3 in *Umod* +/+ (*n* = 4) and *Umod* DEL/+ (*n* = 6) kidneys at 16 weeks with densitometry analysis. Two-tailed *t*-test, Mean ± SD. *p* = 0.0176 (for cleaved CASP9), 0.0012 (cleaved CASP3). **h** Representative IF staining of TUNEL and UMOD (red) with nuclei counterstain (blue) on frozen kidney sections from *Umod* +/+ and *Umod* DEL/+ mice at 24 weeks. The TUNEL-positive cells were quantified. *n* = 5 images/genotype. Red and white arrows: UMOD⁻ and UMOD⁺ cells, respectively. Two-tailed *t*-test, Mean ± SD. ****p* < 0.0001. Source data are provided as a Source Data file.

investigated whether STING knockdown using a lentiviral shRNA-driven approach could dampen cellular inflammation and apoptosis. We tested two shRNAs directed against STING (shSTING). As shown in Fig. 5m, expression of STING was robustly reduced in HEK 293 cells by both sequences compared with that by a scrambled shRNA control (shControl). shSTING 1 was chosen for further studies due to the better knockdown efficacy. shSTING treatment, as compared with shControl, significantly mitigated the increased STING/p-TBK1/p-NF-κB/p-IRF3 expression (Fig. 5n and Supplementary Fig. 8f), as well as augmented inflammatory gene expression (Fig. 5o) in DEL cells. As a result, increased caspase-9 and caspase-3 cleavage in DEL cells was attenuated by STING knockdown (Fig. 5p). Together, these data directly demonstrate that cGAS-STING signaling is activated by cytosolic leak of mtDNA in H177-R185 del cells, which mediates proinflammatory cytokine production and cell death.

## Loss of MANF in TALs deteriorates autophagy suppression and kidney fibrosis in ADTKD

After we successfully generated the knock-in mouse and cell models resembling ADTKD-*UMOD*, we sought to explore the function of MANF in ER-stressed mutant TALs. Double IF staining of MANF and UMOD revealed increased MANF expression distributed within UMOD⁺ tubules, as early as by 3 weeks of age, in the mutant kidneys (Supplementary Fig. 9a). By 16 weeks, MANF upregulation in the TALs expressing the mutant UMOD became conspicuous in whole-kidney sections by staining (Fig. 6a), as well as in whole kidney and TAL protein lysates by WB (Fig. 6b and Supplementary Fig. 9b). We further confirmed increased MANF level in the UMOD⁺ tubules in the kidney biopsy from a patient carrying *UMOD* H177-R185 del (Fig. 6c).

Next, we studied the biological function of MANF in ADTKD-*UMOD*. To specifically ablate MANF from UMOD⁺ TALs in *Manf* fl/fl mice, we used a tamoxifen (TAM)-dependent *Umod*-driven CreER^T2 (estrogen receptor) recombinase line (*Umod* IRES CRE-ERT2, hereafter referred to as *Umod* CE), in which IRES-CRE-ER^T2 was knocked into the 3′ untranslated region (UTR) near the stop codon of the *Umod*. Their offspring, *Umod* DEL/+;*Manf* fl/fl and *Umod* DEL/+;*Umod* CE/+;*Manf* fl/fl, as well as their WT control littermates *Umod* +/+;*Manf* fl/fl and *Umod* +/+;*Umod* CE/+;*Manf* fl/fl, was given TAM for three doses (3 mg/mouse), every other day at 5 weeks of age. Subsequently, the TAM-administered cohorts of mice were followed until 12 weeks. The specific removal of MANF from TAL cells after TAM treatment was confirmed by PCR of genomic DNA isolated from TALs (Supplementary Fig. 9c). As expected[18], the amplified band of 1300 bp contained exon 3 of *Manf* and Frt- and LoxP-sites in *Umod* +/+;*Manf* fl/fl and *Umod* DEL/+;*Manf* fl/fl mice, while 531 bp represented the targeted *Manf* -/- lacking the exon 3 in *Umod* +/+;*Umod* CE/+;*Manf* fl/fl and *Umod* DEL/+;*Umod* CE/+;*Manf* fl/fl littermates (Supplementary Fig. 9c). In line with the PCR results, this manipulation led to a significant reduction of MANF protein levels in the TALs producing mutant UMOD, as evidenced by co-IF staining of MANF and UMOD (Fig. 6d) and immunoblot analysis of MANF (Fig. 6e). At 1 month after administration of TAM, we first observed a significant elevation of BUN

in *Umod* DEL/+;*Umod* CE/+;*Manf* fl/fl mice compared with *Umod* DEL/+;*Manf* fl/fl mice at 9 weeks (Supplementary Fig. 9d and Fig. 6f). At 12 weeks, the accelerated progression of kidney disease in the mutant mice deficient of MANF in the TALs became much more pronounced (Fig. 6f). Consistent with exacerbation of the kidney function after depletion of MANF in TALs, Picrosirius red staining revealed marked increases in collagens I and III in the corticomedullary junction area (Fig. 6g, h), and WB showed an increase in NGAL and FN in *Umod* DEL/+;*Umod* CE/+;*Manf* fl/fl kidneys compared with *Umod* DEL/+;*Manf* fl/fl kidneys at 12 weeks (Fig. 6i and Supplementary Fig. 9e).

Mechanistically, MANF deficiency in TALs was linked to an increased expression of p50 ATF6, on both WT and *Umod* DEL/+ background (Fig. 6j). We also observed that MANF deletion in TALs resulted in a further failure in autophagy, as demonstrated by decreased levels of the positive autophagy regulators p-AMPK (arrow) and FOXO3, as well as increased level of the negative regulator p-mTOR (S2481), and a subsequent increase in the mutant UMOD level in the whole-kidney lysates from *Umod* DEL/+;*Umod* CE/+;*Manf* fl/fl mice vs. *Umod* DEL/+;*Manf* fl/fl mice at 12 weeks (Fig. 6j, k). Together, these data support that MANF upregulation in ER-stressed TAL cells is indispensable for maintaining autophagic activity, and that loss of MANF in mutant TALs exacerbates autophagy inhibition and mutant UMOD accumulation, further intensifying kidney injury in ADTKD-*UMOD*.

## Transgenic (Tg) tubular MANF expression stimulates autophagy and clearance of mutant UMOD in the mouse model of ADTKD-*UMOD*

We next asked whether MANF overexpression can enhance autophagy and play a therapeutic role in ADTKD-*UMOD*. To this end, we generated inducible MANF Tg (TET-MANF) mice, in which the full-length cDNA of human MANF is controlled by a (TetO)7/CMV promoter containing 7 tetracycline-response elements (TREs) (Supplementary Fig. 10a). Next, TET-MANF mice were crossed with Pax8-reverse tetracycline-controlled transcriptional activator (rtTA) mice (Pax8-rtTA mice, abbreviated to P8TA hereafter), in which the pan-tubular specific promoter Pax8 directs expression of rtTA in all renal tubular epithelial cells (Supplementary Fig. 10a). In the presence of tetracycline derivative, doxycycline (DOX), rtTA binds to TRE and initiates transcription of the MANF cDNA in renal epithelial cells (Supplementary Fig. 10a). When TET-MANF was bred to homozygous P8TA/P8TA mice, two genotypes were generated and abbreviated to MANF/P8TA and P8TA. The offspring was induced with DOX at 4 weeks. As shown in Fig. 7a, by 2 weeks after the DOX induction, urinary MANF excretion was easily detected in MANF/P8TA, but not in P8TA mice. Q-PCR of MANF in kidneys using a pair of primers shared in both mouse and human MANF genomic sequences showed a 2.33 ± 0.3 fold of increase of MANF in the DOX-treated bi-transgenic compared with the single Tg mice after 4 weeks administration of DOX (Fig. 7b). At the protein level, double IF staining clearly demonstrated that MANF was induced in both UMOD⁺ tubules (Fig. 7c, red arrows) and UMOD⁻ tubules (Fig. 7c, white arrows) in a perinuclear distribution pattern with 4 weeks of DOX

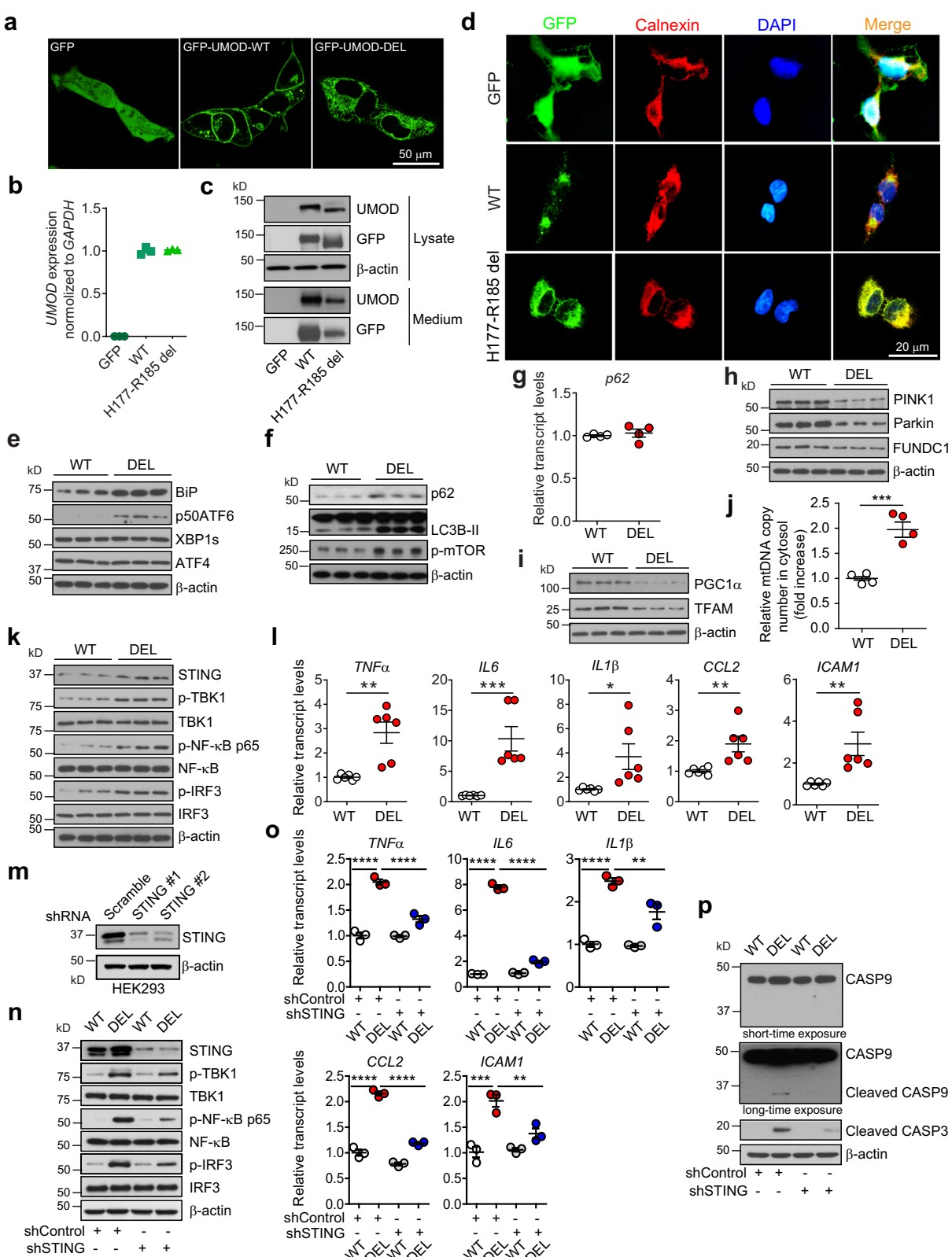

treatment. Co-IF staining of MANF with the proximal tubular marker, lotus tetragonolobus lectin (LTL), another TAL marker, Na-K-Cl cotransporter, and the collecting duct marker, dolichos biflorus agglutinin (DBA) further confirmed that proximal tubules, TALs and collecting ducts all overexpress MANF in DOX-treated MANF/P8TA mice (Supplementary Fig. 10b). In agreement with the staining results, WB showed that the double Tg mice exhibited increased abundance of

MANF in kidneys compared to single Tg mice (Fig. 7d and Supplementary Fig. 11a).

After we had characterized a successful tubular induction of MANF with DOX administration, TET-MANF mice were crossed with $Umod^{DEL/+}$;P8TA/P8TA mice, since TAL-specific rtTA mice are not available. The 4 week-old offspring was started with DOX after the onset of disease. BUN was monitored longitudinally in the offspring

**Fig. 5 | Cytosolic leak of mtDNA activates STING signaling by utilizing stable HEK 293 cell line expressing N-terminal GFP-tagged WT or H177-R185 del UMOD. a** Confocal live imaging showing GFP signal in HEK 293 cells expressing GFP, GFP-tagged WT or mutant H177-R185 del uromodulin. **b** *UMOD* transcript expression assessed by RT-qPCR. Mean ± SD of fold changes of three independent samples from each clone. **c** Cell lysates and media from the indicated cells were analyzed by WBs for expression of GFP-UMOD by using anti-UMOD or anti-GFP antibody. **d** IF analysis of GFP and calnexin with nuclei counterstain (DAPI). **e**, **f**, **h**, i, k Cell lysates of WT and DEL cells were analyzed by WBs for the indicated proteins. *n* = 3/group. **g** Quantitative PCR analysis for mRNA levels of p62, normalized to GAPDH. *n* = 4/group. Mean ± SD. **j** Cytosolic translocation of mtDNA in WT and DEL cells was quantified by q-PCR. The copy number of mtDNA encoding cytochrome *c* oxidase I was normalized to nuclear DNA encoding 18 S ribosomal RNA. *n* = 4/

group. Two-tailed *t*-test, Mean ± SD of fold changes. ***$p = 0.0008$. **l** Quantitative PCR of proinflammatory genes downstream of STING/NF-κB signaling, normalized to GAPDH. *n* = 6/group. Two-tailed *t*-test, Mean ± SD. $p = 0.0019$ (for *TNFα*), 0.0009 (*IL6*), 0.0287 (*IL1β*), 0.0069 (*CCL2*), 0.0068 (*ICAM1*). **m** WB analysis showed knockdown efficacy of STING expression from shSTING1 and shSTING2 vs. a scrambled shRNA control in HEK 293 cells. **n** Cell lysates of WT and DEL cells, treated with shControl or shSTING for 48 h, were analyzed by WBs for the indicated proteins. **o** Quantitative PCR of proinflammatory genes, normalized to GAPDH, in cells treated with shControl or shSTING for 48 h. *n* = 3/group. One-way ANOVA with Tukey's multiple comparisons, Mean ± SD. *TNFα*, *IL6*, *CCL2*: ****$p < 0.0001$; *IL1β*: ****$p < 0.0001$, **$p = 0.0044$; *ICAM1*: ***$p = 0.0003$, **$p = 0.0062$. **p** Cell lysates of WT and DEL cells, treated with shControl or shSTING for 48 h, were analyzed by WBs for cleaved caspases 9 and 3. Source data are provided as a Source Data file.

until 20 weeks. We observed a statistically significant BUN decrease in *Umod*<sup>DEL/+</sup>;MANF/P8TA vs. *Umod*<sup>DEL/+</sup>;P8TA mice from 12 to 20 weeks (Fig. 7e). We further investigated the molecular mechanisms underlying the protective effect of MANF by utilizing the kidney lysates at 14 weeks. First, we confirmed that kidneys from DOX-treated MANF/P8TA mice exhibited significantly increased MANF levels compared to kidneys from DOX-treated P8TA mice, either on the *Umod*<sup>+/+</sup> or *Umod*<sup>DEL/+</sup> background (Fig. 7f and Supplementary Fig. 11b). As MANF is a secreted protein, we further checked total MANF mRNA levels in the isolated TALs at the conclusion of the experiment (20 weeks) by using the pair of primers amplifying both mouse and human MANF genomic sequences. As shown in Fig. 7g, MANF was markedly induced at the transcriptional level in the inducible MANF transgenic mice, on both *Umod*<sup>+/+</sup> and *Umod*<sup>DEL/+</sup> background.

At 14 weeks, ATF6 activation in *Umod*<sup>DEL/+</sup>;P8TA kidneys was substantially suppressed by the tubular upregulation of MANF (Fig. 7h and Supplementary Fig. 11c). Next, we looked at the effect of MANF on autophagic activity. With enhanced expression of MANF in the *Umod*<sup>DEL/+</sup>;MANF/P8TA kidneys, the p62 and LC3B-II levels were decreased compared with those in the *Umod*<sup>DEL/+</sup>;P8TA kidneys, indicating increased autophagic activity (Fig. 7i and Supplementary Fig. 11d). In addition, MANF upregulation in *Umod*<sup>DEL/+</sup>;MANF/P8TA kidneys attenuated expression of p-mTOR (S2481) in *Umod*<sup>DEL/+</sup>;P8TA kidneys, thus promoting autophagic flux (Fig. 7i and Supplementary Fig. 11d). Moreover, increased levels of p-AMPK (arrow) and FOXO3 by MANF overexpression in isolated mutant TALs at 12 weeks (Fig. 7j) corroborated the findings in the whole-kidney lysates at 14 weeks. Transcript analysis of TALs from the indicated genotypes further supports that decreased mRNA levels of key autophagy/mitophagy-related genes, including *Becn1*, *Atg5*, *Atg7*, *Map1lc3b*, *Pink1* and *Bnip3*, in *Umod*<sup>DEL/+</sup>;P8TA TAL cells were restored by MANF upregulation toward WT levels (Fig. 7k). At 20 weeks of age, a marked decrease of mutant UMOD (66.26 ± 4.04% reduction) and p62 in the DOX-treated *Umod*<sup>DEL/+</sup>; MANF/P8TA kidneys was observed compared with those in the DOX-treated *Umod*<sup>DEL/+</sup>;P8TA kidneys (Fig. 7l and Supplementary Fig. 11e). Meanwhile, urinary excretion of UMOD was increased by 92.02 ± 6.35% in the DOX-treated *Umod*<sup>DEL/+</sup>;MANF/P8TA mice compared with DOX-treated *Umod*<sup>DEL/+</sup>;P8TA mice (Fig. 7m and Supplementary Fig. 11f).

To directly demonstrate that the secreted MANF acts as an autophagy activator, we treated the WT and mutant cells producing H177-R185 del UMOD with human recombinant MANF (hrMANF, R&D). The cells were pulsed to low dose (10 μg/mL) brefeldin A (BFA, an inhibitor of protein trafficking from the ER to the Golgi apparatus) for 4 h to promote UMOD retention in the ER. After recovery in the culture media for 20 h, the cells were treated with 5 μg/mL of hrMANF for 24 h. In contrast to DEL cells in the absence of MANF, hrMANF-treated DEL cells manifested a decrease in the abundance of p-mTOR (S2481) and p62, and an increase in p-AMPK and FOXO3 (Fig. 7n and Supplementary Fig. 11g). Due to the increased autophagy flux, hrMANF treatment enhanced autophagic degradation of mutant UMOD in the DEL cell

lysates, and improved secretion of UMOD into the cell culture medium in DEL cells, presumably due to attenuated ER stress (Fig. 7o and Supplementary Fig. 11h). Consequently, treatment with hrMANF abolished activation of caspase-9 and caspase-3 in the mutant cells (Fig. 7p). Collectively, these studies suggest that secreted MANF promotes autophagic activity and removal of misfolded UMOD in ADTKD-*UMOD*.

## MANF overexpression in renal tubules restores mitochondrial homeostasis, suppresses STING activation, and ameliorates kidney injury and fibrosis in ADTKD-*UMOD*

Given that MANF upregulation stimulates expression of p-AMPK, the guardian of mitochondrial homeostasis and metabolism, we therefore reasoned that tubular MANF overexpression may improve mitochondrial function in ADTKD-*UMOD* mouse model. Prompted by increased transcriptional expression of *Pink1* and *Bnip3* by MANF (Fig. 7k), we first investigated the effect of MANF on mitophagy. Increased protein abundance of PINK1/Parkin/BNIP3L (arrow) by MANF overexpression in isolated mutant TAL cells at 12 weeks concurred with the notion that MANF promoted mitophagy (Fig. 8a). In addition, at 12 weeks, WB revealed increased expression of PGC1α in the TALs purified from DOX-treated *Umod*<sup>DEL/+</sup>;MANF/P8TA kidneys compared with TALs from DOX-treated *Umod*<sup>DEL/+</sup>;P8TA kidneys (Fig. 8b and Supplementary Fig. 12a). In alignment with increased PGC1α expression, mutant TALs with MANF overexpression exhibited increased levels of mitochondrial gene transcripts, including *mt-Nd4*, *mt-Co1*, *mt-CytC and mt-Rnr2* (Fig. 8c), and ETC proteins (Fig. 8d and Supplementary Fig. 12b) relative to mutant TALs without MANF transgenic expression at 16 weeks. Finally, the high resolution respirometry showed that mitochondrial respiration was significantly higher in the MANF-overexpressing mutant TALs as compared to mutant TALs at 16 weeks (Fig. 8e).

Correlating with the effect on improving mitochondrial health, tubular upregulation of MANF ameliorated STING activation in *Umod*<sup>DEL/+</sup>;P8TA TALs, as evidenced by decreased protein levels of STING, p-TBK1 and p-NF-κB in *Umod*<sup>DEL/+</sup>;MANF/P8TA TALs at 16 weeks (Fig. 8f and Supplementary Fig. 12c). We further conducted q-PCR analysis of downstream targets of STING/NF-κB signaling on isolated TAL cells at 16 weeks. The mRNA levels confirmed that upregulation of *Il1b*, *Tnfa*, *Il6* and *Icam1* gene expression in mutant UMOD-expressing tubular epithelium was blunted by MANF overexpression (Fig. 8g). Consequently, enhanced expression of MANF in renal tubules repressed expression of the tubular injury marker NGAL, and fibrosis marker FN at 20 weeks (Fig. 8h and Supplementary Fig. 12d). In line with immunoblot results, Trichrome staining further confirmed that MANF overexpression in renal tubules significantly blocked fibrosis in the mutant kidneys (Fig. 8i). In summary, our data have convincingly demonstrated that tubular targeted MANF upregulation promotes mitophagy, improves mitochondrial biogenesis and OXPHOS, and mitigates STING activation, leading to attenuation of tubular injury and fibrosis in ADTKD-*UMOD*.

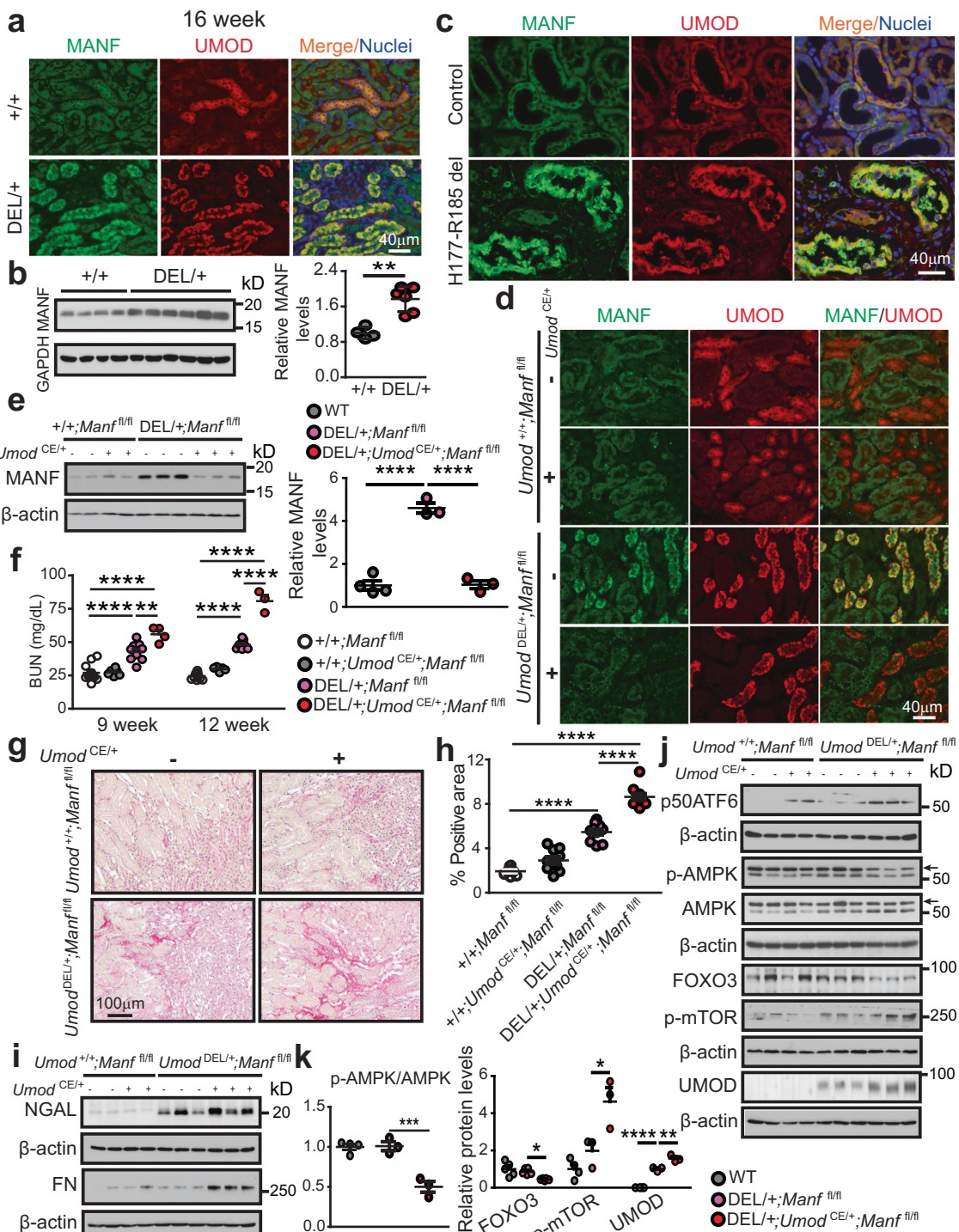

**Fig. 6 | Loss of MANF in TALs deteriorates autophagy suppression and kidney fibrosis in ADTKD-*UMOD*. a** Co-localization of MANF and UMOD with nuclei counterstain (blue) on kidney sections from WT and *Umod*[DEL/+] mice at 16 weeks. **b** WB of MANF in WT (*n* = 4) and *Umod*[DEL/+] (*n* = 6) kidneys at 16 weeks with densitometry analysis. Two-tailed *t*-test, Mean ± SD. **\*\****p* = 0.0011. **c** Kidney sections from a patient with p.H177-R185 del and a normal kidney, stained for MANF and UMOD with nuclei counterstain (blue). **d** Kidney sections from mice of the indicated genotypes were examined by dual IF staining of MANF and UMOD at 9 weeks. **e** Kidney WB of MANF from WT (*n* = 4), DEL/+;*Manf*[fl/fl] (*n* = 3), and DEL/+;*Umod*[CE/+]; *Manf*[fl/fl] (*n* = 3) mice at 12 weeks with densitometry analysis. One-way ANOVA with Tukey's multiple comparisons, Mean ± SD. \*\*\*\**p* < 0.0001. **f** BUN measurements at 9 and 12 weeks. +/+;*Manf*[fl/fl] (*n* = 12, 10 for weeks 9, 12), +/+;*Umod*[CE/+];*Manf*[fl/fl] (*n* = 6, 5 for weeks 9, 12), DEL/+;*Manf*[fl/fl] (*n* = 9, 7 for weeks 9, 12), DEL/+;*Umod*[CE/+];*Manf*[fl/fl]

(*n* = 4, 3 for weeks 9, 12). One-way ANOVA with Tukey's multiple comparisons, Mean ± SD. \*\**p* = 0.0049; \*\*\*\**p* < 0.0001. **g** Kidney sections stained with Picrosirius red and quantified in **h**. *n* = 10 images/genotype. One-way ANOVA with Tukey's multiple comparisons, Mean ± SD. \*\*\*\**p* < 0.0001. **i** Representative WBs of NGAL and FN in WT and DEL/+ kidneys without or with MANF deletion at 12 weeks. **j, k** WB of p50ATF6, p-AMPK/AMPK (arrows), FOXO3, p-mTOR (S2481) and UMOD in WT and DEL/+ kidneys without or with MANF loss in TALs at 12 weeks (**j**) with densitometry analysis (**k**). WT (*n* = 5 for FOXO3, *n* = 4 for other proteins), DEL/+;*Manf*[fl/fl] (*n* = 6 for FOXO3, *n* = 3 for other proteins), DEL/+;*Umod*[CE/+];*Manf*[fl/fl] (*n* = 6 for FOXO3, *n* = 3 for other proteins). One-way ANOVA with Tukey's multiple comparisons, Mean ± SD. \*\*\*\**p* < 0.0001. DEL/+;*Umod*[CE/+];*Manf*[fl/fl] vs DEL/+;*Manf*[fl/fl]: *p* = 0.0008 (for p-AMPK/AMPK), 0.015 (FOXO3), 0.0198 (p-mTOR) and 0.0011 (UMOD). Source data are provided as a Source Data file.

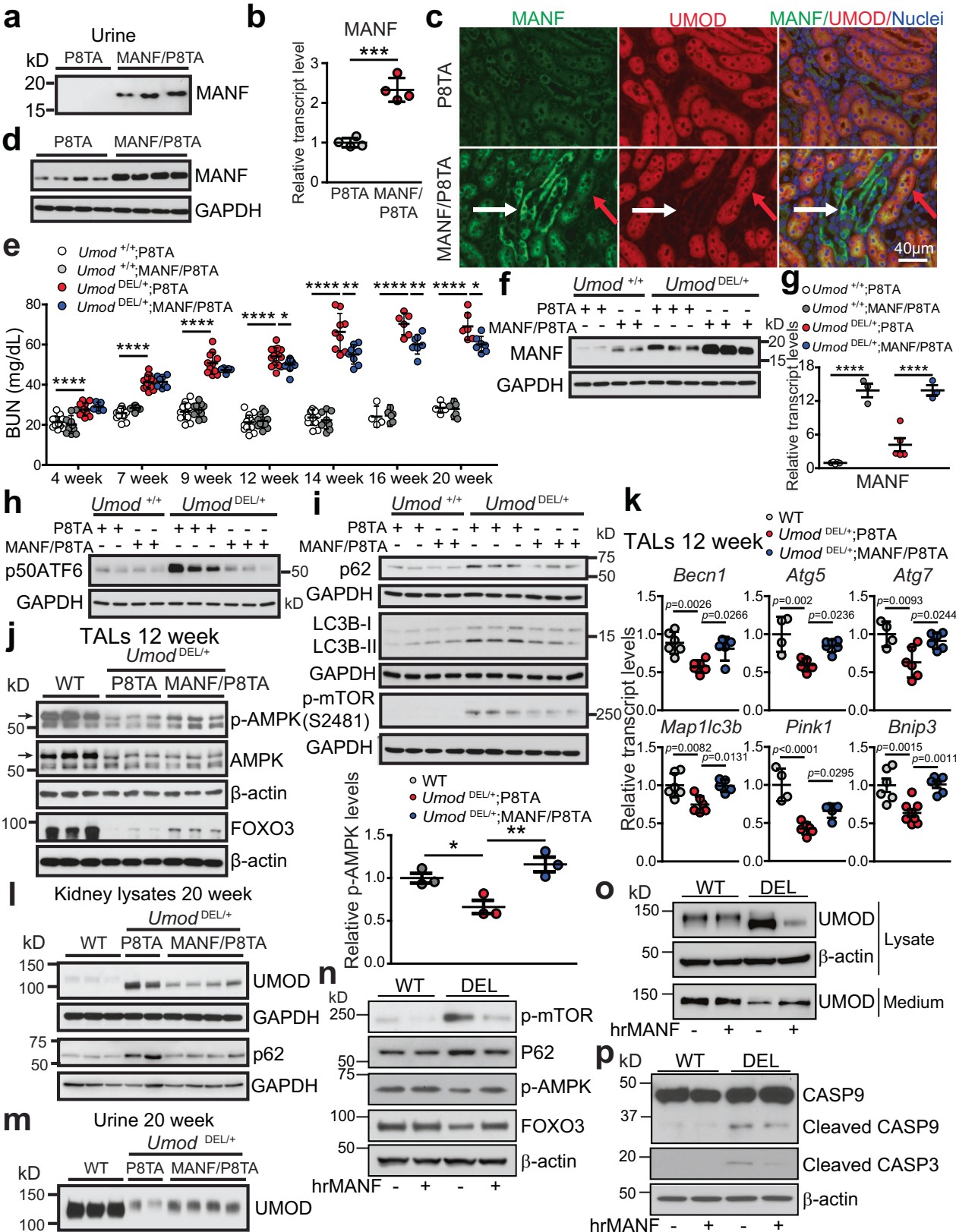

## MANF upregulation in renal tubules slows down CKD progression in *Umod* <sup>C147W/+</sup> mice

To explore whether MANF is also a biotherapeutic protein for ADTKD caused by other *UMOD* mutations, we utilized *Umod* <sup>C147W/+</sup> mice[9], which harbor a human *UMOD* missense mutation and develop CKD and renal fibrosis more slowly than our deletion mutation mice. TET-

MANF were bred with *Umod* <sup>C147W/+</sup>;P8TA/P8TA mice and their offspring, including *Umod* <sup>C147W/+</sup>;P8TA, *Umod* <sup>C147W/+</sup>;MANF/P8TA together with their WT littermates, *Umod* <sup>+/+</sup>;P8TA, *Umod* <sup>+/+</sup>;MANF/P8TA were induced with DOX at 6 weeks, when BUN did not exhibit a significant elevation between the mutants and WT littermates. Serial BUNs were monitored over the 24-week course. As shown in Supplemental

**Fig. 7 | MANF treatment in vivo and in vitro stimulates autophagy in ADTKD. a** WB of urine MANF, indexed to urine creatinine (4 µg), from the indicated DOX-treated mice. **b** Quantitative-PCR of total MANF transcript levels in 4-week DOX-treated P8TA ($n = 4$) and MANF/P8TA ($n = 4$) kidneys. Two-tailed $t$-test, Mean ± SD. ***$p = 0.0002$. **c** IF staining of MANF, UMOD and nuclei counterstain (blue) on the indicated kidney sections after 4 weeks of DOX administration. Red and white arrows point to UMOD$^+$ and UMOD$^-$ tubules, respectively. **d** WB of kidney MANF from the indicated groups with 4-week DOX treatment. **e** BUN measurements over a 20-week period. For weeks 4, 7, 9, 12, 14, 16, 20, $n = 14, 17, 14, 19, 12, 4, 5$ for +/+;P8TA, $n = 10, 10, 10, 13, 8, 6, 6$ for +/+;MANF/P8TA, $n = 11, 17, 11, 18, 9, 6, 6$ for DEL/ + ;P8TA, and $n = 5, 8, 5, 11, 8, 7, 7$ for DEL/ + ;MANF/P8TA. One-way ANOVA with Tukey's multiple comparisons, Mean ± SD. ****$p < 0.0001$, DEL/ + ;MANF/P8TA vs DEL/ + ;P8TA: *$p = 0.033$ (12 weeks), 0.0214 (20 weeks); **$p = 0.0022$ (14 weeks), 0.006 (16 weeks). **f, h, i** Kidney WB of indicated proteins from the indicated genotypes at 14 weeks. **g** Q-PCR of total MANF transcript levels in WT and DEL/+ TALs without ($n = 5$ for both genotypes) or with tubular MANF overexpression ($n = 3$ for both genotypes) at 20 weeks. One-way ANOVA with Tukey's multiple comparisons, Mean ± SD. ****$p < 0.0001$. **j** WBs of the indicated proteins from TALs of the indicated groups ($n = 3$/group) at 12 weeks with densitometry analysis of p-AMPK. One-way ANOVA with Tukey's multiple comparisons, Mean ± SD. *$p = 0.0414$, **$p = 0.0075$. **k** Q-PCR of TALs from the indicated groups at 12 weeks for autophagy and mitophagy-related genes. For WT, DEL/ + ;P8TA and DEL/ + ;MANF/P8TA, $n = 6$, 6, 5 for *Becn1*, *Map1lc3b*, $n = 4$, 5, 6 for *Atg5*, $n = 4$, 6, 6 for *Atg7*, $n = 4$, 6, 5 for *Pink1*, $n = 6$, 8, 6 for *Bnip3*. One-way ANOVA with Tukey's multiple comparisons, Mean ± SD. **l, m** WBs of UMOD and P62 from kidneys (**l**) or urines (**m**) of the indicated groups at 20 weeks. The urinary UMOD excretion was indexed to urine creatinine (4 µg). **n–p** Cell lysates and media of stable WT and DEL cells, which were pulsed with BFA, and then treated without or with hrMANF, were analyzed by WBs for the indicated proteins. Source data are provided as a Source Data file.

Fig. 13a, BUN became significantly lower in the DOX-treated *Umod*$^{C147W/+}$;MANF/P8TA mice compared to DOX-treated *Umod*$^{C147W/+}$;P8TA mice since 10 weeks of age. Light microscopic examination of Picrosirius red-stained kidney sections revealed less renal fibrosis at 24 weeks (Supplementary Fig. 13b) in the *Umod*$^{C147W/+}$; MANF/P8TA kidneys vs. *Umod*$^{C147W/+}$;P8TA kidneys at 24 weeks. Together, the data obtained in another line of *Umod* mutant mice support that the renal protective effect of MANF may not be restricted to an individual *Umod* mutation. Further mechanistic studies will be performed to determine whether MANF also promotes autophagy/mitophagy and protects mitochondrial function in *Umod*$^{C147W/+}$ mice.

## Discussion

In summary, by generating a mouse model that resembles ADTKD-*UMOD* carrying a prevalent human mutation, and by employing both loss- and gain-of-function studies of MANF, our work has discovered that MANF is an important regulator of autophagy/mitophagy and mitochondrial homeostasis in mutant TALs through activation of p-AMPK in ADTKD-*UMOD*, thereby mitigating cGAS/STING activation and promoting autophagic degradation of mutant UMOD (Fig. 9). Our study uncovers mechanisms of MANF action and provides a foundation for pharmacological strategies that target MANF, which may have profound therapeutic potential to ameliorate kidney fibrosis and slow down kidney function decline in ADTKD-*UMOD*.

ADTKD is a genetically heterogeneous disease, caused by mutations in *UMOD*, *MUC1* (mucin 1), *HNF1B* (hepatocyte nuclear factor 1β), *REN* (renin) and *SEC61A1* (α1 subunit of translocon 61). Of all 135 *UMOD* mutations identified so far, with the exception of 6 in-frame deletions, most are missense mutations and ~50% of known *UMOD* mutations affect cysteine leading to protein misfolding[35]. In the current study, we generated a *Umod* in-frame deletion mouse carrying the most common deletion mutation in human patients. Although cysteine is not present in the deleted 9 amino acids, YETLTEYWR (p.Tyr178-Arg186 del), the UMOD deletion mutation activates ER stress, as other missense mutations, presumably due to protein misfolding and aggregation. Notably, different UPR branches are involved in the different *Umod* mutation mouse models. For example, ATF4 and XBP1s are activated (ATF6 was not checked) in *Umod*$^{C147W/+}$ mice[9], whereas these two branches are not upregulated in our deletion mutation mice. It is also interesting to note that in a recently generated ADTKD-*MUC1* mouse model that recapitulates a frame shift mutation in *Muc1* (*Muc1*-fs mice), ATF6 is the prominent UPR branch that regulates ER stress response in response to the MUC-1 protein aggregates while with minimal transcriptional changes in the other two branches[36]. The function of ATF6 can be adaptive or maladaptive, depending on the different contexts, and few studies have been published regarding the role of ATF6 in the kidney disease. It has been shown that overexpression of p50ATF6 in a proximal tubular cell line decreases the expression of peroxisome proliferator-activated receptor-α (PPAR-α),

the master regulator of lipid metabolism, to repress β-oxidation of fatty acid, and thus to cause excessive lipid droplet formation. In contrast, ATF6$^{-/-}$ mice have sustained expression of PPAR-α and less tubular lipid accumulation following ischemia-reperfusion injury, leading to less renal fibrosis[37]. That augmented kidney expression of p50ATF6 through MANF deletion and attenuated expression of ATF6 via MANF overexpression are associated with the opposite kidney functional outcomes in our mouse model might suggest that the effect of ATF6 activation is deleterious in chronic ER-stressed TAL cells. Whether ATF6 directly regulates AMPK activity and thus autophagy/mitophagy, and whether reprogramming the ER proteostasis environment through ATF6 manipulation can impact the disease outcome in ADTKD-*UMOD* are important questions that remain to be answered.

Our study has demonstrated suppressed autophagy in the mutant TALs in ADTKD-*UMOD*. The molecular mechanism whereby autophagy deficiency leads to accumulation of mutant UMOD needs further investigation. The autophagy receptor p62 has an N-terminal Phox and Bem1p (PB1) domain, which mediates self-oligomerization, and a C-terminal ubiquitin-associated (UBA) domain capable of interaction with polyubiquitinated proteins. Meanwhile, p62 can bind to LC3 through its LIR (LC3-interacting region)/LRS (LC3 recognition sequence), which enables sequestration of the ubiquitinated proteins and p62 in the autophagosomes. This process is achieved efficiently by self-oligomerization of p62[38]. We will perform further studies to elucidate whether p62 directly binds to polyubiquitinated UMOD and whether ER stress in the mutant TALs, besides its impact on p-AMPK activity, may induce upregulation of certain proteins, which can interfere with p62 binding to the ubiquitinated UMOD or LC3, thus obstructing autophagic flux.

A functional mitophagy system acts as a scavenger of damaged mitochondria and thereby maintains mitochondrial homeostasis. To enable formation of mitophagosomes in mitophagy, mitochondrial priming is regulated by both ubiquitin-dependent PINK1/Parkin pathway and ubiquitin-independent mitophagy receptors, including BNIP3/BNIP3L/FUNDC1[25]. It has been reported that PINK1, Parkin or BNIP3 deficiency exacerbates ischemic/reperfusion-, cisplatin-, contrast medium-, or sepsis-induced acute kidney injury (AKI)[39–43], as well as unilateral ureter obstruction-induced CKD[44,45]. Concurring with the notion that mitophagy plays a protective role in proximal tubular cells under pathogenic conditions, our study further illuminates the important therapeutic implication of MANF as a mitophagy/autophagy activator in ADTKD-*UMOD*, which can also directly promote autophagic clearance of mutant UMOD. Mechanistically, MANF exerts its mitophagy/autophagy-promoting effect by enhancing p-AMPK-FOXO3 signaling. Recently, we have identified that MANF binds to and antagonizes cell surface receptor neuroplastin to modulate its anti-inflammatory effect[46]. More studies will be required to understand whether neuroplastin is also involved in regulating AMPK and mitophagy/autophagy in ADTKD.

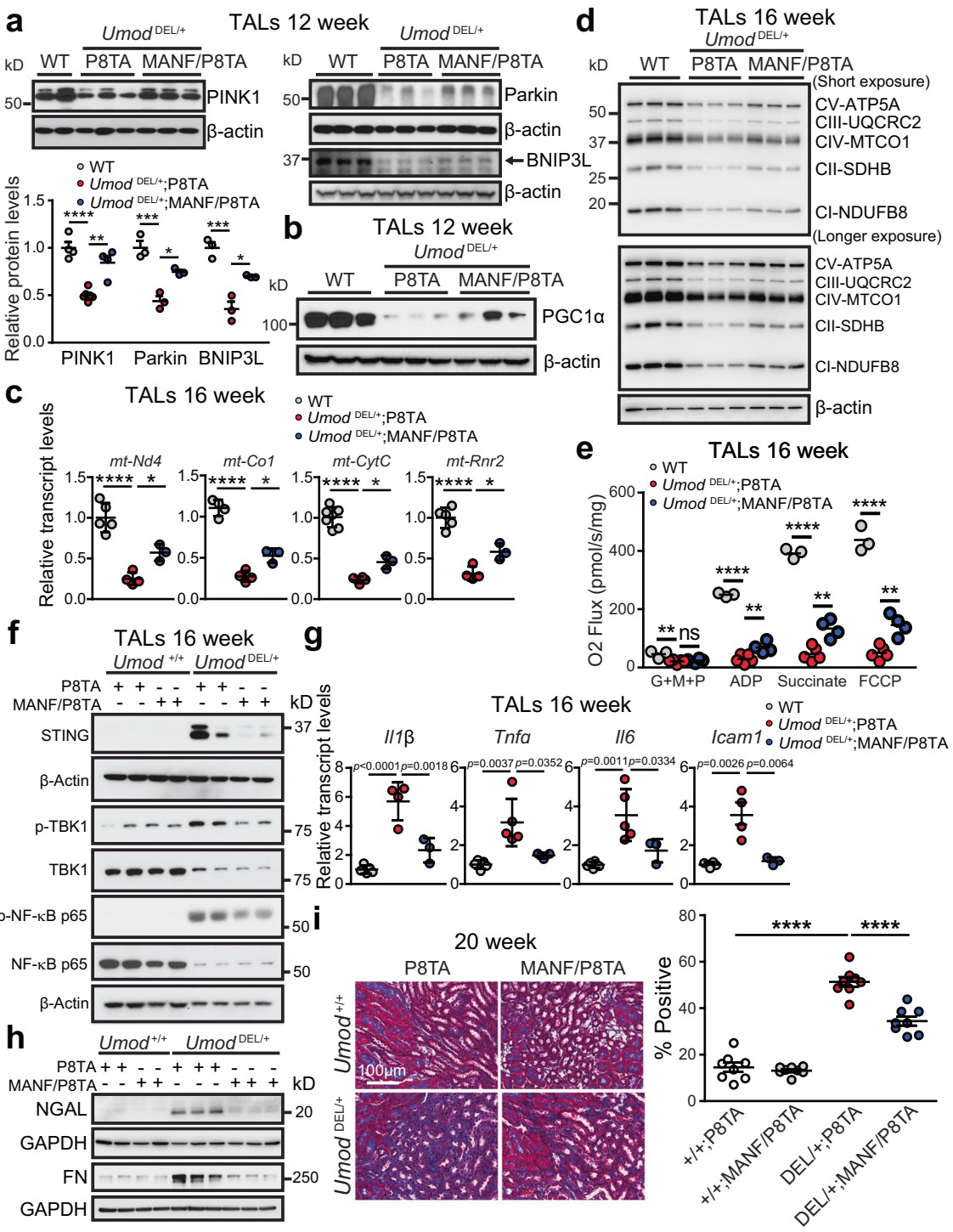

Our study also provides evidence that disrupted mitochondrial integrity due to impaired mitophagy and mitochondrial biogenesis in mutant TALs contributes to STING activation and renal fibrosis in ADTKD-*UMOD*. In alignment with recent studies that mitochondrial damage causes STING activation, renal inflammation and tubular cell death in CKD[32] and AKI[33], our study highlights the importance of STING signaling in ADTKD-*UMOD*. Furthermore, tubular MANF overexpression directly suppresses STING activation in the mutant TALs.

To date, there is no disease-specific therapy to treat or halt the disease progression of ADTKD until the patients reach end-stage renal disease. It has been shown that blocking TNFα-mediated inflammation

by using a soluble TNF receptor inhibitor slows disease progression in *Umod*[C147W/+] mice[9]. Given the hierarchical order starting from mitochondrial quality control failure for the eventual activation of multiple proinflammatory cytokines in our deletion mutation mice, we envision that treatment with the upstream p-AMPK enhancer MANF, which can restore homeostasis of dysfunctional mitochondria and inhibit STING activation, would be more effective in blocking activation of a myriad of inflammatory genes, including TNFα, in ADTKD-*UMOD*. Furthermore, anti-TNF treatment would not enhance autophagic clearance of the mutant UMOD protein, which is most likely the root cause of the disease. MANF is an 18kD protein and recombinant MANF is readily available. In the future, we will continue to test the therapeutic

**Fig. 8 | Renal tubular upregulation of MANF promotes mitophagy and improves mitochondrial biogenesis, leading to abrogation of STING activation and kidney fibrosis in ADTKD-*UMOD*. a** WBs of PINK1, Parkin and BNIP3L (arrow) in TALs isolate from the kidneys of the indicated groups at 12 weeks with densitometry analysis. For WT, DEL/+ ;P8TA, DEL/+ ;MANF/P8TA, *n* = 4, 6, 4 for PINK1, *n* = 3, 3, 3 for Parkin and BNIP3L. One-way ANOVA with Tukey's multiple comparisons, Mean ± SD. DEL/+ ;P8TA vs WT: *p* < 0.0001 (PINK1), 0.0007 (Parkin) and 0.0005 (BNIP3L). DEL/+ ;MANF/P8TA vs DEL/+ ;P8TA: *p* = 0.0011 (for PINK1), 0.0141 (Parkin) and 0.0125 (BNIP3L). **b** WB of PGC1α in TALs from the indicated groups at 12 weeks. **c** Transcript analysis of *mt-Nd4, mt-Co1, mt-CytC* and *mt-Rnr2* from TALs of WT (*n* = 4–7), DEL/+ ;P8TA (*n* = 4) and DEL/+ ;MANF/P8TA (*n* = 3) at 16 weeks. One-way ANOVA with Tukey's multiple comparisons, Mean ± SD. ****$p$ < 0.0001. DEL/+ ;MANF/P8TA vs DEL/+ ;P8TA: *$p$ = 0.031 (for *mt-Nd4*), 0.0115 (*mt-Co1*), 0.0357 (*mt-CytC*) and 0.0248 (*mt-Rnr2*). **d** WBs of ETC expression in TALs of the indicated genotypes at 16 weeks. **e** Assessment of mitochondrial function

using an OROBOROS Oxygraph system in permeabilized TALs of WT (*n* = 3), DEL/+ ;P8TA (*n* = 5) and DEL/+ ;MANF/P8TA (*n* = 4) at 16 weeks following sequential additions of G + M + P; ADP; succinate and FCCP. One-way ANOVA with Tukey's multiple comparisons, Mean ± SD. DEL/+ ;P8TA vs WT: **$p$ = 0.0085, ****$p$ < 0.0001. DEL/+;MANF/P8TA vs DEL/+ ;P8TA: *p* = 0.8880 (ns, G + M + P), 0.0095 (ADP), 0.0017 (Succinate), 0.0052 (FCCP). **f** WBs of STING signaling in the TALs of the indicated genotypes at 16 weeks. **g** Transcript analysis of inflammatory genes from TALs of WT (*n* = 4-6), DEL/+ ;P8TA (*n* = 4 for *Il1b, Icam1, n* = 5 for *Tnfa, Il6*) and DEL/+ ;MANF/P8TA (*n* = 3) at 16 weeks. One-way ANOVA with Tukey's multiple comparisons, Mean ± SD. **h** WBs of NGAL and FN in the kidneys of the indicated genotypes at 20 weeks. **i** Trichrome staining of kidney sections from the indicated groups at 20 weeks with quantification. *n* = 8 images/genotype. One-way ANOVA with Tukey's multiple comparisons, Mean ± SD. ****$p$ < 0.0001. Source data are provided as a Source Data file.

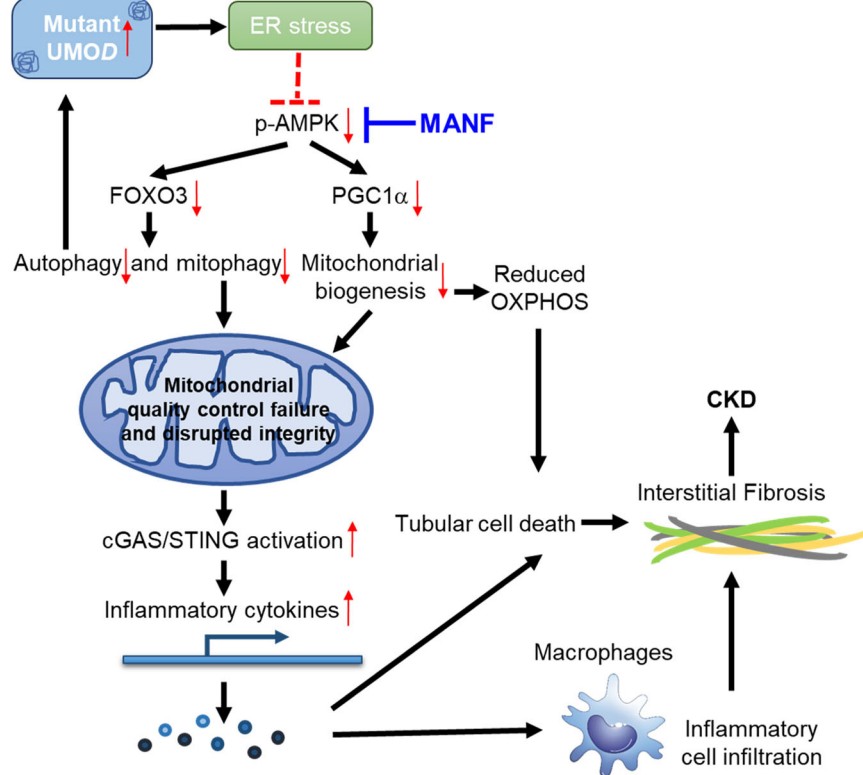

**Fig. 9 | Modulation of kidney fibrosis in ADTKD-*UMOD* by MANF.** Mutant UMOD triggers ER stress response, which may directly or indirectly inhibit AMPK activation. Tubular overexpression of MANF after the onset of ADTKD promotes autophagy/mitophagy via activation of p-AMPK-FOXO3 signaling, and enhances mitochondrial biogenesis via p-AMPK-PGC1α axis. Moreover, elimination of dysfunctional mitochondria and restoration of mitochondrial homeostasis through

increased tubular MANF expression abrogate STING activation and mitigate tubular pathological inflammation. Finally, increased autophagic clearance of mutant UMOD by tubular MANF upregulation would further stop the malicious cycle between impaired autophagy and ER stress intensification, eventually protecting kidney function in ADTKD-*UMOD*.

application of MANF as an effective strategy for the treatment of ADTKD patients caused by various gene mutations. In addition, whether MANF can treat other proteinopathies resulting from mutant protein aggregates and altered proteostasis, such as Alzheimer's disease[47], amyotrophic lateral sclerosis[2] and retinitis pigmentosa[48], is of great interest for the future investigation.

## Methods

All research described here complies with all relevant ethical regulations and all animal research protocols, which have been approved by the Institutional Animal Care and Use Committee at Washington University School of Medicine.

### Antibodies and primers (Supplementary Tables 1–4)

**shRNAs and reagents.** Lentiviral vector pLKO.1 was used for generating shRNAs targeting STING. Target sequences were selected as follows: shSTING#1, 5′-GCATTACAACAACCTGCTACG-3′; shSTING#2, 5′-GCATCAAGGATCGGGTTTACA-3′.

BFA was purchased from Sigma-Aldrich (St Louis, MO, CAT B6542), hrMANF protein was from R&D (Minneapolis, MN, CAT 3748-MN-050), TAM was from Sigma-Aldrich (CAT T5648) and doxycyline food (1500 ppm, irradiated) was from El Mel (St. Charles, MO, CAT 1814935). Biotinylated LTL was from Vector Laboratories (CAT B-1325-2), Biotinylated DBA was from Vector Laboratories (CAT B-1035-5) and CM-H$_2$DCFDA was from Invitrogen (CAT C6827).

**Animals.** Animal experiments conformed to the National Institutes of Health Guide for the Care and Use of Laboratory Animals were approved by the Washington University Animal Studies Committee (animal protocol number 20-0286). Mice were maintained on a 12 h light/dark cycle at 20–24 °C and controlled humidity (30–70%, usually around 50%) in an AAALAC accredited facility. Both male and female littermates ranging in age from 3–24 weeks were used for experiments. Sex was not tracked as a biological variable, as no difference related to the kidney disease phenotype was observed between the male and female littermates. The distribution of sex was equal across all groups. Littermate controls were used in all experiments. TAM was given to *Umod* DEL/+;*Manf* fl/fl and *Umod* DEL/+;*Umod* CE/+;*Manf* fl/fl mice, as well as their WT control littermates *Umod* +/+;*Manf* fl/fl and *Umod* +/+;*Umod* CE/+; *Manf* fl/fl at 5 weeks of age, 3 mg per oral gavage on each mouse, every other day for three doses. Doxycyline food (1500 ppm, irradiated) was given to *Umod* DEL/+;P8TA and *Umod* DEL/+;MANF/P8TA mice, as well as their WT control littermates *Umod* +/+;P8TA and *Umod* +/+;MANF/P8TA mice at 4 weeks of age from 4–20 weeks.

Mice were observed closely for the signs of a premorbid state. These signs include hypoactivity; shallow, rapid, and/or labored breathing; failure to groom; failure to respond to stimuli; hunched posture; dehydration, and weight loss. Once these signs appeared, mice were euthanatized. All surviving mice were euthanatized at the end of experiment. Methods of euthanasia are consistent with the recommendations of the Panel on Euthanasia of the American Veterinary Medical Association. Mice were anaesthetized by intraperitoneal injection of ketamine/xylazine, followed by cervical dislocation.

**Generation of UMOD Y178-R186 deletion mutant mice.** *Umod* DEL/+ mice were generated by Washington University Diabetes Research Center Transgenic & ES Cell Core and the Washington University Mouse Genetics Core. CRISPR gRNAs for in vitro testing were identified using CRISPOR (http://crispor.tefor.net/) and synthesized as gBlocks (IDT). In vitro target specific gRNA cleavage activity was validated by transfecting N2A cells with PCR amplified gRNA gblock and Cas9 plasmid DNA (px330, Addgene) using ROCHE Xtremegene HP. Cell pools were harvested 48 h later for genomic DNA prep, followed by Sanger sequencing of PCR products spanning the gRNA/Cas9 cleavage site and TIDE analysis (https://tide.nki.nl/) of sequence trace files. CRISPR sgRNA and Cas9 protein for injection were purchased from IDT and complexed to generate the ribonucleoprotein (RNP) for injection along with a 200 nucleotide single-stranded oligodeoxynucleotide (ssODN) donor DNA (IDT). Injection concentrations were: 50 ng/µl Cas9, 20 ng/µl gRNA and 20 ng/µl ssODN. C57BL/6 J F1 mice at 3–4 weeks of age (The Jackson Laboratory, Bar Harbor, ME) were superovulated by intraperitoneal injection of 5 IU pregnant mare serum gonadotropin (SIGMA-Aldrich), followed 48 h later by intraperitoneal injection of 5 IU human chorionic gonadotropin (SIGMA-Aldrich). Mouse zygotes were obtained by mating C57BL/6 J males with superovulated C57BL/6 J females at a 1:1 ratio. One-cell fertilized embryos were injected into the pronucleus and cytoplasm of each zygote. Founder genotyping was done by deep sequencing (MiSeq, Ilumina). Mosaic founders were crossed to C57BL/6 J (The Jackson Laboratory) to generate heterozygous F1 offspring. F1 offspring was deep sequenced to confirm correctly targeted alleles. gRNA sequence: 5′-GCTGCGCCAGTACTCAGTCA-3′. ssODN sequence: 5′- GCACACAGGTCTCAGCCATGCGAACGCCACCCTGGCCTGTGAACCG GTACCAGCCGTGCAGACCCGCGTCACAGGAGTAGCCCACACCATACTC TGTGCTTGTATTGCAGGGGTCTTGACACACCAGCTTTCCATCCGGGCC CTGGGGCAAGCAGTCCAGTCCTGGCTCACAGGAGCCTGGGGAGCACT CACAGTACCAA-3′.

**Generation of inducible TET-MANF Tg mice.** TET-MANF mice were generated by Dr. Maria Lindahl. In brief, to produce these transgenic mice, the full-length cDNA of human MANF was inserted into a *Tet-op-mp1* vector[49] (a kind gift of Prof. Harold E. Varmus, Weill Cornell Medicine, NY and Prof. Katerina Politi, Yale University, New Haven, CT) between a seven direct repeats of the tet-operator sequence (Tet-O$_7$) and the mouse protamine 1 poly (A) and intron sequence. The transgenic construct was injected into the pronucleus of fertilized egg cells of FVB/N strain at the Gene Modified Mouse Unit of Laboratory Animal Center, University of Helsinki. Four positive TET-MANF transgenic mouse founders were obtained and transgene copy numbers were analyzed by Southern blotting and quantitative RT-PCR.

*Umod* C147W/+ mice on the C57BL/6 background were published before[9].

*Manf* fl/fl on the C57BL/6 background was generated by Dr. Maria Lindahl[18].

*Umod* IRES CRE-ERT2 on the C57BL/6 background was generated by Dr. Andrew P. McMahon in the University of South California and purchased from The Jackson Laboratory (Stock No: 030601). In the CRISPR/Cas9 generated *Umod* IRES CRE-ERT2 mice, IRES-CRE-ER$^{T2}$ was knocked into the 3′ UTR near the stop codon of the *Umod*. *Umod* CE/+ or *Umod* CE/CE mice do not exhibit any phenotype abnormality for up to 2 years.

Pax8-rtTA mice on the C57BL/6 background were purchased from The Jackson Laboratory (Stock No: 007176). The hemizygous transgenic mice were intercrossed to generate the homozygous transgenic mice for some breedings by utilizing q-PCR. Forward: 5′- TCGATTCT-CATGCCCTTCGC-3′, reverse: 5′-GAGCGAGTTTCCTTGTCGTC-3′.

**Construct vectors of GFP-UMOD-WT and GFP-UMOD-H177-R185 del**
pLenti-CMV-Blast-DEST was obtained from Addgene (#17451) and the UMOD gene was synthesized by Genscript (Piscataway, NJ) and provided in a cloning vector. pLenti-CMV-Blast-SS-eGFP-UMOD was cloned using overlap PCR and Gateway cloning. Here, the eGFP sequence was amplified with a forward primer containing an attB site, signal sequence, and complementary region of eGFP. A reverse primer contained a region complementary to eGFP and a UMOD overhang. The UMOD gene was amplified with a forward primer containing a region complementary to UMOD and an eGFP overhang. The reverse primer contained an attB site and complementary region to UMOD. The eGFP and UMOD genes were then amplified with the respective primers, and then overlap PCR was performed. The amplified product was then inserted into pDONR 221 using a Gateway BP reaction. The donor plasmid, pDONR221-SS-eGFP-UMOD, was then inserted into pLenti-CMV-Blast-DEST via an LR reaction to yield pLenti-CMV-Blast-SS-eGFP-UMOD. The H177-R185 deletion was constructed with a similar approach, with a primer annealing to the respective region to yield the alternative construct. Cloning was then confirmed via Sanger sequencing. Please note that eGFP was referred to as GFP in the text and figures.

**Lentiviral transduction of HEK 293 cells**
Human embryonic kidney (HEK) 293 (ATCC, CAT CRL-1573) and HEK 293 T (ATCC, CAT CRL-3216) cells were cultured at 37 °C with 5% CO$_2$ in Dulbecco's modified Eagle medium (DMEM) (Gibco, Carlsbad, USA) containing 2 mmol/L L-glutamine, 100 U/mL penicillin, 100 µg/mL streptomycin, and 10% (vol/vol) fetal bovine serum (Sigma-Aldrich). Lentiviruses were produced by co-transfection of HEK 293 T cells with the packaging vector psPAX2 (Addgene #12260), the envelope expression plasmid pMD2.G (Addgene #12259) and lentiviral vector expressing GFP-UMOD-WT or GFP-UMOD-H177-R185 del at a ratio of 3:1:4 using Lipofectamine 3000 Reagent (ThermoFisher Scientific, Waltham, MA) following the manufacturer's instructions. The medium was changed 8 h post-transfection. The virus-containing supernatant was harvested and passed through a 0.45-µm filter 72 h post-transfection. The lentivirus stock expressing GFP-UMOD-WT or GFP-UMOD-H177-R185 del was used for transduction. After HEK 293 cells were infected by lentiviruses, media were changed 24 h post-

transduction and cells were selected with 4 µg/mL Blasticidin (Sigma-Aldrich, CAT 15205) for 3 days after transduction.

## Immunofluorescence staining

For dual staining of MANF or BiP with UMOD in mouse kidneys, 4% paraformaldehyde-fixed paraffin-embedded sections were used. After dewaxing, kidney sections were subjected to antigen retrieval by immersion in 10 mM citric acid buffer (pH 6.0) for 5 min at 95 °C. Nonspecific binding was blocked by incubating kidney sections with 1% BSA for 30 min at room temperature. The slides were stained with a rabbit anti-mouse BiP antibody (Proteintech) or a rabbit anti-mouse (human) MANF antibody (Abnova) together with a rat anti-mouse UMOD antibody (R&D) overnight at 4 °C, followed by Alexa 488 or 594-conjugated anti-rabbit or anti-rat secondary antibodies.

For co-staining of MANF with biotinylated LTL or DBA in mouse kidneys, paraffin-embedded sections were used. After dewaxing, nonspecific avidin binding was blocked by an avidin/biotin blocking kit (Vector laboratories) before incubating kidney sections with 1% BSA for 30 min at room temperature. The slides were stained with a rabbit anti-mouse MANF antibody (Abnova) together with biotinylated LTL (Vector Laboratories) or DBA (Vector Laboratories) overnight at 4 °C, followed by an Alexa 488-conjugated anti-rabbit secondary antibody and Alexa 594-conjugated streptavidin, as well as Hoechst 33342.

For double staining of MANF with NKCC2 in mouse kidneys, frozen sections were employed without antigen retrieval. Nonspecific binding was blocked by incubating kidney sections with 1% BSA for 1 h at room temperature. The slides were stained with a goat anti-human MANF antibody (R&D) and a rabbit anti-rat (mouse) NKCC2 antibody (Alpha Diagnostic) together overnight at 4 °C, followed by Alexa 594-conjugated anti-goat or 488-conjugated anti-rabbit secondary antibodies.

Formalin-fixed, paraffin-embedded sections of human kidneys underwent the same process of deparaffinization, antigen retrieval and blocking. The slides were incubated with the same rabbit anti-human (mouse) MANF antibody (Abnova), or a rabbit anti-human p62 antibody (Cell Signaling) in combination with a mouse IgG2b antibody against human uromodulin (RayBiotech). A biotinylated anti-mouse antibody was used to amplify fluorescence signals for uromodulin, followed by the corresponding Alexa 488-conjugated anti-rabbit secondary antibody to detect MANF or p62, and by Alexa 594-labeled streptavidin to detect uromodulin. The nuclei were counterstained with Hoechst 33342.

For double staining of FN (Abcam), SMA (Sigma-Aldrich) or F4/80 (Invitrogen) with UMOD in mouse kidneys, frozen sections were employed without antigen retrieval. Ten randomly selected fields (magnification, x 400) in each kidney were evaluated. TUNEL analysis was performed with the In Situ Cell Death Detection Kit, Fluorescein (Ver16.0, Roche, Mannheim, Germany) per the instructions.

For staining of CM-H$_2$DCF in mouse kidneys, frozen sections were employed. Nonspecific binding was blocked by incubating kidney sections with 1% BSA for 1 h at room temperature. The slides were stained with 5 µM CM-H$_2$DCFDA (Invitrogen) for 30 min at 37 °C, followed by nuclear staining of DAPI for 30 min at room temperature.

For immunocytochemistry, the stably transduced HEK 293 cells were seeded on coverslips to grow for 48 h. The cells were fixed in 4% paraformaldehyde for 10 min, permeabilized in 0.1% triton for 10 min, and blocked with 1% BSA for 1 h at room temperature, followed by incubation with mouse anti-calnexin antibody overnight at 4 °C. The coverslips were washed with PBS and incubated with Alexa 594-conjugated anti-mouse secondary antibody for 1 h at 37 °C. The nuclei were counterstained with DAPI.

After immunofluorescence staining, images were captured under a fluorescence microscope (Nikon 80i), and analyzed using cellSens standard software (V1.18). Positively stained area was quantified using Image J software (NIH, V1.5.3).

## Light microscopy, electron microscopy and confocal microscopy

For light microscopy, mouse kidneys were fixed in 4% PFA, dehydrated through graded ethanols, embedded in paraffin, sectioned at 5 µm, and stained with H&E, Masson's Trichrome or Picrosirius red by the Musculoskeletal Research Center Morphology Core. For TEM, tissues were fixed, embedded in plastic, sectioned, and stained by the Washington University Electron Microscope Facilities. For live cell imaging, Images were captured using a Nikon Eclipse Ti-E inverted confocal microscope equipped with 10× Plan Fluor (0.30 NA), 20× Plan Apo air (0.75 NA), 60 × Plan Fluor oil immersion (1.4 NA), or 100× Plan Fluor oil immersion (1.45 NA) objectives (Nikon). Images were processed and analyzed using Elements AR 5.21 (Nikon).

Fibrosis severity, as quantified by Masson's Trichrome, was scored in the kidney corticomedullary junction area at × 400 magnification using a counting grid with 140 intersections. The number of grid intersections overlying Trichrome-positive (blue) interstitial areas was counted and expressed as a percentage of all grid intersections. For this calculation, intersections that were in tubular lumen and glomeruli were subtracted from the total number of grid intersections[50].

## Purification of mouse primary TAL epithelial cells

Mouse UMOD-producing epithelial cells were purified by preparing single cell digests from mouse kidneys of the different genotypes[9]. Briefly, kidneys were decapsulated, minced, and then incubated in a shaking water bath (180 rpm) at 37 °C for 60 min with Type 1 Collagenase (2 mg/mL; Sigma-Aldrich) and DNase (100 U/mL; Sigma-Aldrich) in serum-free DMEM/F12 (Sigma-Aldrich) with vortexing every 15 min. After incubation, enzymatic digestion was halted by addition of 1 volume of DMEM/F12 with 10% serum. The tissue digest was mixed thoroughly and filtered (40 µm) to get the single cell suspension. UMOD-producing cells were separated from the single cell suspension using the Dynabead magnetic bead conjugation system, whereby biotinylated, sheep anti-UMOD monoclonal antibody (R&D) was incubated with Dynabeads Biotin Binder (Thermo Fisher, CAT 11047). For each kidney isolate, 100 µL beads were conjugated with 4 µg anti-UMOD antibody at 4 °C for 45 min. Beads were then washed twice with 1% BSA/PBS buffer, and incubated with the single cell suspension in a final volume of 1 mL at 4 °C for 60 min. Beads and UMOD-positive cells were then trapped against a magnet, and residual UMOD-negative cells were washed away. UMOD-positive fractions were divided for RNA and protein analysis.

## RNA sequencing and analysis

Library preparation was performed with 500 ng of total RNA where the ribosomal RNA was removed by an RNase-H method using RiboErase kits (Kapa Biosystems, Wilmington, MA, CAT KK8561), fragmented in reverse transcriptase buffer and heated to 94 °C for 8 min, and then reverse transcribed to yield cDNA using SuperScript III RT enzyme (Life Technologies, Grand Island, NY, CAT 18080044, per manufacturer's instructions) and random hexamers. A second strand reaction was performed to yield ds-cDNA. cDNA was then blunt ended, had an A base added to the 3' ends, and then had Illumina sequencing adapters ligated to the ends. Ligated fragments were then amplified for 12–15 cycles using primers incorporating unique dual index tags, and the indexed fragments for each sample were then pooled in an equimolar ratio and sequenced on an Illumina NovaSeq-6000 using paired end reads extending 150 bases.

Basecalls and demultiplexing were performed with Illumina's bcl2fastq software (V2.0) and a custom python demultiplexing program with a maximum of one mismatch in the indexing read. RNA-seq reads were then aligned to the Ensembl release 76 primary assembly with STAR version 2.5.1a[51]. Gene counts were derived from the number of uniquely aligned unambiguous reads by Subread:featureCount

version 1.4.6-p5[52]. All gene counts were then imported into the R/Bioconductor package EdgeR version 3.32.1[53] and TMM normalization size factors were calculated to adjust for samples for differences in library size. Ribosomal genes and genes not expressed in at least 5 samples greater than one count-per-million were excluded from further analysis. The TMM size factors and the matrix of counts were then imported into the R/Bioconductor package Limma version 3.46.0[54]. Weighted likelihoods based on the observed mean-variance relationship of every gene and sample were then calculated for all samples with the voomWithQualityWeights[55]. Differential expression analysis was then performed to analyze for differences between conditions, and the results were filtered for only those genes with Benjamini-Hochberg false-discovery rate (FDR) adjusted $p$ values less than or equal to 0.05.

For each contrast extracted with Limma, global perturbations in known Gene Ontology (GO) terms and KEGG pathways were detected using the R/Bioconductor package GAGE version 2.40.2[56] to test for changes in expression of the log2 fold-changes reported by Limma in each term versus the background log2 fold-changes of all genes found outside the respective term. GO terms and KEGG pathways with Benjamini-Hochberg adjusted $p$ values less than 0.05 were considered statistically significant. The R/Bioconductor package heatmap3[57] was used to display heatmaps across groups of samples for each GO with a Benjamini-Hochberg FDR adjusted $p$ value less than or equal to 0.05.

## Western blot analysis

The isolated UMOD-positive or -negative epithelium and stable HEK 293 cells were lysed by RIPA buffer (Cell signaling, CAT 9806) containing protease and phosphatase inhibitor cocktails (Roche). The mouse kidneys were extracted using the same lysis buffer with the protease and phosphatase inhibitors and homogenized by sonication. The protein concentrations of cell and kidney lysates were determined by Bio-Rad protein assay (Hercules, CA) using BSA as a standard. Urine volume from individual mouse was normalized to urine creatinine excretion. Denatured proteins were separated on SDS polyacrylamide gel electrophoresis (SDS-PAGE) and then transferred to PVDF membranes. Proteins were not heat denatured for detection of mitochondrial OXPHOS proteins. Blots were blocked with 5% non-fat milk for 1 h at room temperature, and then incubated overnight with primary antibodies at 4 °C. The membranes were washed with Tris-buffered saline/Tween buffer and incubated with the appropriate HRP–conjugated secondary antibodies. The proteins were then visualized in an x-ray developer using ECL plus (GE, Pittsburgh, PA) or Clarity Max ECL (Biorad) detection reagents or Chemidoc MP imaging instrument (Biorad). To ensure equal protein loading, the same blot was stripped with stripping buffer (25 mM glycine + 1% SDS, pH = 2.0) and then incubated with a HRP-conjugated anti-mouse (human) β-actin antibody (Sigma-Aldrich) or anti-mouse (human) GAPDH antibody (Cell Signaling, Danvers, MA). Relative intensities of protein bands were quantified using Image J (NIH, V1.5.3) analysis software.

## mRNA quantification by Real-Time PCR

Total RNA from TALs, stably transduced HEK 293 cells or whole kidneys was extracted using the RNeasy kit (Qiagen, Valencia, CA) with subsequent DNase I treatment. Cellular or kidney RNA was then reverse-transcribed using an RT-PCR kit (Superscript III; Invitrogen). Gene expression was evaluated by quantitative real-time PCR. One μl of cDNA was added to SYBR Green PCR Master Mix (Qiagen) and subjected to PCR amplification (one cycle at 95 °C for 20 s, 40 cycles at 95 °C for 1 s, and 60 °C for 20 s) in a QuantStudio ™ 6 Flex Fast Real-Time PCR System (Life Technologies). Q-PCR was conducted in triplicate for each sample.

## Preclinical PET/ CT imaging

Radiolabeled ⁶⁸Ga-Galuminox was synthesized. ⁶⁸Gallium was eluted from a generator (IGG100-50M) using 0.1 M HCl (1.1 mL), NaOAc buffer (pH 5, 400 μL) was added to the eluent mixture, the pH was adjusted to 4.5, mixed with a solution of the luminol-glycine-NOTA ligand (50 μg, 50 μL) dissolved in a solution of 90% ethanol/10% polyethylene glycol 200, and heated at 100 °C for 15 min. The reaction was monitored using radio-thin layer chromatography. Following completion of the reaction, the reaction mixture was passed through cation exchange column (Phenomenex; Strata-X-C 33 m Polymeric Strong Cation; 30 mg/mL) to remove trace metal impurities and unreacted ⁶⁸Gallium. Finally, the purified radiolabeled fraction was diluted into sterile saline containing 2% ethanol (pH 7) and deployed for PET imaging. Imaging studies were performed in *Umod* ⁺/⁺ and *Umod* ᴰᴱᴸ/⁺ mice at 16 weeks. Both sexes were used. For these studies, mice were anesthetized with isoflurane (2%) via an induction chamber and maintained with a nose cone. After anesthetization, the mice were secured in a supine position and placed in an acrylic imaging tray. Following intravenous tail-vein administration of ⁶⁸Ga-Galuminox (100 μL; 100 μCi; 2% ethanol in saline, 3.7 MBq), static preclinical PET scans were performed over 30–50 min, using Inveon PET/CT scanner (Siemens Medical Solutions). PET data were stored in list mode, and reconstruction was performed using a three-dimensional ordered-subset expectation maximization (3D-OSEM) method with detector efficiency, decay, dead time, attenuation, and scatter corrections applied. For anatomical visualization, PET images were also co-registered with CT images from an Inveon PET/CT scanner. Regions of interests were drawn over the kidney, and standard uptake values (SUVs) were calculated as the mean radioactivity per injected dose per animal weight.

## Biodistribution studies

Following micro PET/CT imaging, mice ($n = 8$) were euthanized by cervical dislocation. Blood samples were obtained by cardiac puncture, kidneys harvested rapidly, and all tissue samples analyzed for γ-activity using Beckman Gamma 8000 counter. All samples were decay-corrected to the time, and the γ-counter was started. Standard samples were counted with the kidneys for each animal and represented 1% of the injected dose. An additional dose was diluted into milliQ water (100 mL) and aliquots (1 mL) were counted with each mouse. Data were quantified as the percentage injected dose (% ID) per gram of tissue (tissue kBq (injected kBq)$^{-1}$ (g tissue)$^{-1}$ ×100).

## Preparation of permeabilized UMOD⁺ and UMOD⁻ cells and high-resolution respirometry

Isolated UMOD-positive (TAL) and UMOD-negative cells were resuspended in cold mitochondrial respiration solution (MIR05, 0.5 mM EGTA, 3 mM MgCl₂, 60 mM K-lactobionate, 20 mM taurin, 10 mM KH₂PO4, 20 mM HEPES, 110 mM sucrose and 1 g/L BSA, pH 7.1) and placed in the Oxygraph 2 K (OROBOROS Instruments, Innsbruck, Austria) for high resolution-respirometry measurements[58–60]. To facilitate uptake of respiratory substrates, cellular plasma membranes were permeabilized in the chamber with digitonin (10 ug/mL) for 20 min. Following permeabilization, routine oxygen consumption was measured by the sequential addition of the following substrates: malate (0.5 mM), glutamate (10 mM) and pyruvate (5 mM) to assess complex I-mediated leak respiration, adenosine diphosphate (ADP, 5 mM) to assess maximal complex I respiration, and succinate (10 mM) to measure complex I and II-mediated respiration. The uncoupling agent FCCP (Carbonyl cyanide p-trifluoro-methoxyphenyl hydrazone, 0.5 μM, titrated 3X) was then added to determine maximal potential electron transport system (ETS) capacity. A period of stabilization followed the addition of each substrate. Following assay completion, cells were recovered from the instrument, assayed for total protein content, and then results were normalized to protein amount and analyzed using the DatLab Software V7.4 (OROBOROS Instruments).

## Mitochondrial fractionation

Stably transduced WT and DEL cells were washed with cold PBS first. Mitochondrial and cytosol fractions were isolated from cells using

Mitochondrial Isolation Kit for Cultured Cells (Thermo Fisher Scientific) according to the manufacturer's protocols.

## mtDNA release assay

DNA was isolated from 200 µL of the cytosolic fraction using a DNeasy Blood & Tissue Kit (Qiagen). Quantitative PCR was employed to measure mtDNA using Applied Biosystems SYBR Green Master Mixes (Agilent Technologies) in a QuantStudio ™ 6 Flex Fast Real-Time PCR System (Life Technologies, software V1.7.1). The copy number of mtDNA encoding cytochrome c oxidase I was normalized to nuclear DNA encoding 18 S ribosomal RNA. The following primers were used: cytochrome c oxidase I (F: 5′-GAGCCTCCGTAGACCTAACC-3′ and R: 5′-TGAGGTTGCGGTCTGTTAGT-3′) and 18 S rRNA (F: 5′-CTCAA-CACGGGAAACCTCAC-3′ and R: 5′- CGCTCCACCAACTAAGAACG-3′).

## BUN measurement

BUN was measured by using a QuantiChrom™ urea assay kit (DIUR-100) (BioAssay Systems, Hayward, CA).

## Urinalysis

Mouse urines were collected by manual restraint. The mouse urines were centrifuged at 1800 g for 10 min to remove debris before being frozen at −70 °C. Urinary Cr concentration was quantified by a QuantiChrom™ creatinine assay kit (DICT-500) (BioAssay Systems).

## Statistics and reproducibility

Statistical analyses were performed using GraphPad Prism (version 8.0) software. Data were expressed as mean ± SD or plots. A 2-tailed Student's $t$ test was used to compare two groups. One-way ANOVA with post-hoc Tukey test was used to compare multiple groups. ns, not significant; $*p < 0.05$; $**p < 0.01$; $***p < 0.001$; $****p < 0.0001$. All experiments were repeated at least two to three times independently with similar results.

## Study approval

All human paraffin-embedded slides from ADTKD patients were obtained from the Wake Forest Cohort under the protocol approved by the Institutional Review Board of Wake Forest University School of Medicine. The written informed consent was obtained from the patients. The slides are stored and provided for the current study in a de-identified manner.

## Reporting summary

Further information on research design is available in the Nature Portfolio Reporting Summary linked to this article.

## Data availability

All data needed to evaluate the conclusions in the paper are present in the paper and/or the Supplementary Materials. The generated raw fastq sequencing files and gene counts data in this study needed for reproducibility with publicly available tools cited in the Methods have been deposited in Gene Expression Omnibus [GEO] database, accession no. GSE230005. Source data are provided with this paper.

## Code availability

The analysis code for the RNA-seq data uses all published and publicly available tools and algorithms as cited in the Methods and only requires familiarity with command line interfaces in a Linux equivalent operating system and the R statistical computing language. All code to reproduce the RNA-seq results is maintained by the Washington University Genome Technology Access Center at The McDonnell Genome Institute (GTAC@MGI) and is under restricted access due to the fee for service design of the core facility. Please email the GTAC@MGI bioinformatics at GTAC-Bioinformatics@path.wustl.edu for access to code and docker computing environments containing the RNA-seq

tools cited in the Methods, which can be made available within 48 h upon request after a 24 h response time during normal business hours.

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

## Acknowledgements

We thank the Washington University Diabetes Research Center Trans-genic & ES Cell Core (supported by NIH P30DK020579) and Mouse Genetics Core for generating the *Umod* ^DEL/+ mice, Musculoskeletal Research Center Morphology Core for histology (supported by NIH P30AR057235), and Electron Microscope Facilities for TEM imaging. We thank Rashmi Nanjundappa, Tao Cheng and Moe Mahjoub for help with confocal live cell imaging. Mice were housed in the Washington Uni-versity Mouse Genetics Core. E.T. is supported by a NCI Cancer Center Support Grant #P30 CA91842 to the Siteman Cancer Center and by ICTS/CTSA Grant# UL1TR002345 from the National Center for Research Resources (NCRR). T.A.P is supported by Cellular and Molecular Biology Core in the Nutrition Obesity Research Center (P30-DK56341). S.K. was

supported by the project LTAUSA19068 from the Ministry of Education, Youth and Sports of the Czech Republic and NU21-07-00033 from the Ministry of Health of the Czech Republic, by the National Institute for Treatment of Metabolic and Cardiovascular Diseases (CarDia; LX22NPO5104) from the Ministry of Education, Youth and Sports of the Czech Republic, and by the project TN02000132/National Center for New Methods of Diagnosis, Monitoring, Treatment and Prevention of Genetic Diseases. A.J.B. was supported by NIH grant R21DK106584. M.E.J. is supported by R35GM128772. F.U. is supported by NIH grants R01 DK112921 and UH2 TR002065. V.S. is supported by grants R0l HL111163, R01 HL142297 and NIBIB P41 EB025815. M.L. is supported by Juvenile Diabetes Research Foundation 17-2013-410 and 2-SRA-2018-496-A-B, grants from the Finnish Diabetes Research Foundation and Grants 117044 and 333974 from the Academy of Finland. Y.M.C. is supported by NIH grants R01 DK105056A1, R21DK131557A1, R03DK106451 and K08DK089015, the Office of the Assistant Secretary of Defense for Health Affairs through the Peer Reviewed Medical Research Program under Award W81XWH-19-1-0320, George M. O'Brien Kidney Research Core Centers (NU GoKidney, NIH P30 DK114857; UAB/UCSD, NIH P30 DK079337), The Washington University Office of Vice Chancellor for Research (OVCR) Seed Grant, Mallinckrodt Challenge Grant, Washington University Center for Drug Discovery Investigator Matching Micro Grant and Faculty Scholar Award from the Children's Discovery Institute of Washington University and St. Louis Children's Hospital. Y.M.C. is a member of Washington University Institute of Clinical and Translational Sciences (UL1 TR000448) and Diabetes Research Center (NIH P30DK020579). We thank the Slim Health Foundation and the Black-Brogan Foundation for support. We also thank the Preclinical Imaging Facility at Mallinckrodt Institute of Radiology for help in performing PE/CT imaging studies. The Preclinical Imaging Facility is supported by the Siteman Cancer Center Support Grant (P30CA091842).

## Author contributions

Y.K. and C.L. designed and performed experiments. C.J.G. designed and performed all experiments in stably transduced HEK 293 cells. Y.L.F. performed experiments. E.T. performed RNA-seq data analysis. A.P. and M.E.J. designed and performed cloning of GFP-UMOD-WT and GFP-UMOD-H177-R185 del. TAP conducted high resolution respirometry. J.S. and V.S. designed and performed PET/CT imaging. K.K., S.K., and A.B. coordinated and supplied patient materials. S.J.P. performed experiments. B.G.J and J.S.D. provided Umod C147W mice. F.U. provided advice in conceiving and conducting experiments. M.L. generated and provided TET-MANF and Manf fl/fl mice. Y.M.C. conceived, designed and supervised the study, and wrote the manuscript. All authors contributed to the review and approval of the manuscript.

## Competing interests

Y.M.C., S.J.P., Y.K., and F.U. are inventors on a patent entitled "Compositions and methods for treating and preventing endoplasmic reticulum (ER) stress-mediated kidney diseases" (US 11,129,871), which was issued by US Patent and Trademark Office in Sep. 2021. Y.M.C. and Y.K. are inventors on a patent entitled "methods of detecting biomarkers of endoplasmic reticulum (ER) stress-associated kidney diseases" (US 10,156,564), which was issued by US Patent and Trademark Office on Dec. 18, 2018. J.S. and V.S. are inventors on a non-provisional patent, entitled "PET tracers for non-invasive imaging of ROS activity" filed by Washington University in St. Louis, St. Louis, MO. Authors (J.S. & V.S.) declare no competing interests. The remaining authors declare no competing interests.

## Additional information

¹Division of Nephrology, Department of Medicine, Washington University School of Medicine, St. Louis, MO, USA. ²Genome Technology Access Center, McDonnell Genome Institute, Washington University School of Medicine, St. Louis, MO, USA. ³Department of Chemistry, Washington University, St. Louis, MO, USA. ⁴Nutrition and Geriatrics Division, Washington University School of Medicine, St. Louis, MO, USA. ⁵Mallinckrodt Institute of Radiology, Washington University School of Medicine, St. Louis, MO, USA. ⁶Section of Nephrology, Wake Forest University School of Medicine, Winston-Salem, NC, USA. ⁷Research Unit of Rare Diseases, Department of Pediatric and Inherited Metabolic Disorders, First Faculty of Medicine, Charles University, Prague, Czech Republic. ⁸Pfizer Worldwide Research and Development, Inflammation & Immunology, Cambridge, MA, USA. ⁹Prime Medicine, Inc, Cambridge, MA, USA. ¹⁰Division of Endocrinology, Metabolism, and Lipid Research, Department of Medicine, Washington University School of Medicine, St. Louis, MO, USA. ¹¹Department of Neurology, Washington University School of Medicine, St. Louis, MO, USA. ¹²Department of Biomedical Engineering, School of Engineering & Applied Science, Washington University, St. Louis, MO, USA. ¹³Institute of Biotechnology, HiLIFE, University of Helsinki, Helsinki, Finland. ¹⁴Department of Cell Biology & Physiology, Washington University School of Medicine, St. Louis, MO, USA. ¹⁵These authors contributed equally: Yeawon Kim, Chuang Li. ✉e-mail: ychen32@wustl.edu

