## [Peer Review File · Nature Communications]

REVIEWER COMMENTS

Reviewer #1 (Remarks to the Author):

In this manuscript Kim et al. investigated the role of mesencephalic astrocyte-derived neurotrophic factor (MANF) in autosomal dominant tubulo-interstitial kidney disease due to uromodulin mutation (ADTKD-UMOD). To accomplish this goal, they generated a number of mutant mice including conditional, Tamoxifen and Doxycycline-inducible mice using various molecular techniques, including CRISPR/Cas9. In addition, at times they used mutant thick ascending limb (TAL) and HEK 293 cell lines. Umod DEL/+ mice conceivably recapitulated the biological events seen in ADTKD-UMOD patients. These included ER stress, decreased p-AMPK, perturbations in autophagy, mitophagy, mitochondrial biogenesis, reduced mitochondrial respiration, increased generation of inflammatory cytokines and oxidant stress, mtDNA leakage, activation of cGAS/STING, tubular dysfunctions, tubulo-interstitial fibrosis and a rise in the renal physiological parameters. All these pathobiological processes were well characterized in this manuscript that were relevant to the mutation of uromodulin. Some of these perturbations were reciprocated in mutant TALs. These changes were normalized in double mutant mice (uromodulin + MANF) overexpressing MANF, especially the interstitial fibrosis – the hallmark of ADTKD-UMOD. On the other hand, these pathological changes worsened in mice with loss of endogenous MANF. From these data the authors concluded that administration MANF by correcting the perturbations in pAMPK and ER stress modulates the series of downstream events leading to the amelioration of aberrant tubular homeostasis and interstitial fibrosis.

This is a seminar study highlighting the role of MANF in the amelioration of interstitial fibrosis and abnormal renal physiological parameters, and thus implying its potential use as a biotherapeutic agent. Although, the study is very exhaustive it has been carried in very diligent manner. The data are convincing. The conclusions drawn in this study are well-supported by the data. The subject matter is suitable for Nature Communications' audience.

Reviewer #2 (Remarks to the Author):

In the present study, the authors created a mouse model that mimics ADTKD-UMOD and linked ER stress and accumulation of the protein to impaired autophagy (which has been reported in another mouse model) and mitochondrial dysfunction (which also has been described before), linking it now to cGAS/STING activation and downstream inflammation and fibrosis. The authors present data and propose a very novel finding, that MANF over-expression can rescue the phenotype by stimulating

autophagy/mitophagy and help clearing of the accumulated misfolded protein. These findings could have a great impact on the field and lead to therapies.

Overall, this is quite an impressive study, and the authors need to be congratulated on this massive work, which included generation of multiple mouse models, crossing many mouse line, innovative in vivo imaging of mitochondrial dysfunction and oxidative stress, isolation of TAL cells and performing a myriad of functional and analytical assays, and human relevance. I do believe that this work will add important advances in the field, will be highly impactful.

The strengths of this study were mentioned above. Other noteworthy positive important findings:

- 1- A clear mechanism linking ER stress to inflammation through impaired mitophagy, leaking of mtDNA, activation of STING.. etc
- 2- Work of isolated TAL cells, which removed a lot of noise
- 3- use of innovative technique for mitochondrial dysfunction and oxidative stress.
- 4- Use of various in vivo models various crossing of transgenics, which likely have required a lot of patience and resources.
- 5- demonstration that MANF over expression in kidney tubules could be beneficial even in another model of ADTKD-UMOD

Overall, it is obvious that the authors went at great length to demonstrate, as best as they could, what they are proposing.

There are few major limitations, which the authors should address/ discuss clearly and/or acknowledge in the limitations:

- 1- In general, quantitation was not performed on all the blots presented, which limits interpretation.
- 2- the cell culture system using transfection of HEK cells has significant limitations:
 - It is unclear how much wild Uromodulin goes to the surface (Fig5D, also co-localizes with calnexin).
 - Also, brefeldin A has to be used to augment the injury, which can introduce by itself changes and non-specific noise.
 - The position of the fused GFP (which terminus?) is not specified and could in itself produce stress.
 - The effect of recombinant MANF does not appear to be significant on p62, p-AMPK, FOXO3 (7M) and unclear if statistically significant on the readouts.

---- Is the Umod release in the medium WT or mutant? shouldn't there be difference in molecular weight that reflect the transfection product?

3- MANF targeted deletion: does this lead to more Umod retention?

4- Manf overexpression:

---what happens to Umod levels quantitatively in the kidney and urine? Manf was already upregulated in DEL/+. How much more over expression did the authors achieve in the TAL? how could they separate from a confounding effect of a protective effect from expression in other tubular segments?

---In figure 7E-H, MANF overexpression cancels activation of ATF6, whereas the suppression of UN elevation and p62 accumulation is partial. Any thoughts about the difference? Does this mean that MANF can inhibit only a part of the signaling cascade caused by ADTKD-induced ER stress?

5- The authors demonstrated that MANF overexpression is also effective for ADTKD caused by another mutation, Umod C147W/+ mice, a previously reported autophagy- impaired mice. However, Umod C147W/+ mice are reported to decrease LC3B protein expression, whereas Umod DEL/+ mice in this study increase LC3B expression. What does the difference mean and how do the authors reconcile their different interpretation of autophagy markers?

6- Figure S2, XBP1 appears to be increased?

minor comments:

1-Same blots of GAPDH and beta actin appear to be replicated in several figures.

2- would be informative to have a supplemental data sheet with list of TAL genes differentially altered by mutation (gene name, fold change, adjusted p-value..etc)

3- In Figure 2A, ATF6 is upregulated 12 weeks ADTKD-UMOD mice. How about 3-week-old ADTKD-UMOD mice, which accumulates the under-glycosylated form of mutant UMOD "due to ER retention" in Figure 1D?

4- In Figure 3H, the authors showed overproduction of mitochondrial ROS in ADTKD-UMOD mice using quantitative PET/CT molecular imaging. What age of mice were used? Could decreased renal excretion

affect the results if they are older than 4 weeks and impairs renal function? Have you quantified ROS in other methods?

5- In figure 3K a and b, the scale bars are missing.

6- In figure 5P, DEL shControl increases cleaved caspase as well as full-length caspase 9. Is it possible that total caspase-9 expression is increased?

7- In figure 7H, does MANF treatment also correct Umod DEL/+ -induced LC3B upregulation?

8- The molecular mechanism of autophagy of mutant Umod should be mentioned at least in the discussion part. Is the mutant UMOD ubiquitinated? Do mutant UMOD directly bind p62?

9- More details on the production of the mice, particularly when generating a mouse with distinct transgenic mutations of the Umod gene on separate chromosomes (Del/Cre-ET)

10- consider toning down the "toxic proteinopathy" in the title, since what is presented applies mostly to UMOD

Reviewer #3 (Remarks to the Author):

This study generated a mouse model that resembles ADTKD-UMOD carrying a prevalent human mutation, and employed role of MANF in ADTKD-UMOD. The work discovered that MANF was an important regulator of autophagy/mitophagy and mitochondrial homeostasis of mutant TALs through activation of p-AMPK in ADTKD-UMOD, thereby mitigating cGAS/STING activation and promoting autophagic degradation of mutant UMOD. However, there are several concerns found in this manuscript. The concerns listed below should be addressed.

Specific comments:

1. The title of this article named "MANF stimulates autophagy and restores mitochondrial homeostasis to treat toxic proteinopathy". However, the study only investigated one of the toxic proteinopathy, ADTKD-UMOD, which not match the title. Adding the results about other kinds of toxic proteinopathy or replacing the title would be better.

2. The study used two kinds of animal models about MANF, including MANF deletion in mutant TALs and MANF overexpression in renal tubules mouse models. Why delete or overexpress MANF in different

kidney cells? The author should discuss the reasons and provide the expression of MANF in different kidney cells under physiological and pathological status.

3. Figure 4C showed highlighted kidney ingress of massive inflammatory cells and few kidney cyst. This work used PAS staining, H&E staining might reveal the inflammatory infiltration better.

4. Result 6 showed TAL cell-specific MANF knockout caused autophagy failure and renal fibrosis. How about the mitochondrial function, renal injury and inflammation after MANF specific knockout in TAL cell?

5. In Figure 6B, 6D and 7D, the author showed IF staining of MANF, UMOD and Hoechst 33342 on kidney paraffin sections. IF staining of MANF and TAL cell marker or tubules marker would better explain the specific knockout or overexpress MANF in TAL cells or tubules. The quantification of MANF expression in TAL cells and tubules should be helpful.

Reviewer #4 (Remarks to the Author):

Kim et al provided the function of MANF in the regulation of organelle homeostasis through detailed experiments. Authors demonstrated that tubular overexpression of MANF after the onset of disease stimulates autophagy/mitophagy and clearance of the misfolded UMOD, and promotes mitochondrial biogenesis through p-AMPK enhancement. In contrast, genetic ablation of endogenous MANF upregulated in the mutant mouse and human tubular cells worsens autophagy suppression and kidney fibrosis. The results of this paper are very interesting, but the author's group used the same method to generate *Umod*(C147W/+) KI mice, and various analyses were performed. The pathway impairment was also pointed out, and it was reported that inactivation of FOXO3 via p-AKT there is involved, and that TNF-alpha pathway is activated (J Clin Invest. 2017 Nov 1;127(11):3954-3969.). In light of this, the paper is very interesting but lacks novelty. Analysis of MANF in ADTKD patients is also needed.

Reviewer #1:

In this manuscript Kim et al. investigated the role of mesencephalic astrocyte-derived neurotrophic factor (MANF) in autosomal dominant tubulo-interstitial kidney disease due to uromodulin mutation (ADTKD-UMOD). To accomplish this goal, they generated a number of mutant mice including conditional, Tamoxifen and Doxycycline-inducible mice using various molecular techniques, including CRISPR/Cas9. In addition, at times they used mutant thick ascending limb (TAL) and HEK 293 cell lines. Umod DEL/+ mice conceivably recapitulated the biological events seen in ADTKD-UMOD patients. These included ER stress, decreased p-AMPK, perturbations in autophagy, mitophagy, mitochondrial biogenesis, reduced mitochondrial respiration, increased generation of inflammatory cytokines and oxidant stress, mtDNA leakage, activation of cGAS/STING, tubular dysfunctions, tubulo-interstitial fibrosis and a rise in the renal physiological parameters. All these pathobiological processes were well characterized in this manuscript that were relevant to the mutation of uromodulin. Some of these perturbations were reciprocated in mutant TALs. These changes were normalized in double mutant mice (uromodulin + MANF) overexpressing MANF, especially the interstitial fibrosis – the hallmark of ADTKD-UMOD. On the other hand, these pathological changes worsened in mice with loss of endogenous MANF. From these data the authors concluded that administration MANF by correcting the perturbations in pAMPK and ER stress modulates the series of downstream events leading to the amelioration of aberrant tubular homeostasis and interstitial fibrosis.

This is a seminar study highlighting the role of MANF in the amelioration of interstitial fibrosis and abnormal renal physiological parameters, and thus implying its potential use as a biotherapeutic agent. Although, the study is very exhaustive it has been carried in very diligent manner. The data are convincing. The conclusions drawn in this study are well-supported by the data. The subject matter is suitable for Nature Communications' audience.

We appreciate very much the positive comments from the Reviewer 1.

Reviewer #2:

In the present study, the authors created a mouse model that mimics ADTKD-UMOD and linked ER stress and accumulation of the protein to impaired autophagy (which has been reported in another mouse model) and mitochondrial dysfunction (which also has been described before), linking it now to cGAS/STING activation and downstream inflammation and fibrosis. The authors present data and propose a very novel finding, that MANF over-expression can rescue the phenotype by stimulating autophagy/mitophagy and help clearing of the accumulated misfolded protein. These findings could have a great impact on the field and lead to therapies.

Overall, this is quite an impressive study, and the authors need to be congratulated on this massive work, which included generation of multiple mouse models, crossing many mouse line, innovative in vivo imaging of mitochondrial dysfunction and oxidative stress, isolation of TAL cells and performing a myriad of functional and analytical assays, and human relevance. I do believe that this work will add important advances in the field, will be highly impactful.

The strengths of this study were mentioned above. Other noteworthy positive important findings: 1- A clear mechanism linking ER stress to inflammation through impaired mitophagy, leaking of mtDNA, activation of STING etc. 2- Work of isolated TAL cells, which removed a lot of noise. 3- use of innovative technique for mitochondrial dysfunction and oxidative stress. 4- Use of various in vivo models various crossing of transgenics, which likely have required a lot of patience and resources. 5- demonstration that MANF over expression in kidney tubules could be beneficial even in another model of ADTKD-UMOD.

Overall, it is obvious that the authors went at great length to demonstrate, as best as they could, what they are proposing.

There are few major limitations, which the authors should address/ discuss clearly and/or acknowledge in the limitations:

We are very grateful for the supportive comments from the Reviewer 2, who has appreciated the value, the amount of work, and the time (about 4 years) that we have spent in this project. We have done additional experiments to address the very thoughtful questions from the Reviewer 2.

1- In general, quantitation was not performed on all the blots presented, which limits interpretation.

Due to the limited space in each figure, we had not provided all densitometry data. Per the suggestion, we have pooled more blots and done densitometry on all WBs, except for very few blots, in which the difference between two groups is absence vs. presence. All the quantification data are included in **Supplementary Figs. 5, 8, 9e, 11 and 12** in addition to a few added densitometry graphs next to the corresponding blots if space allows in the figures.

2- the cell culture system using transfection of HEK cells has significant limitations:

---It is unclear how much wild Uromodulin goes to the surface (Fig5D, also co-localizes with calnexin).

In **Fig. 5d**, we used 0.1% triton to permeabilize the cell membrane after fixing the cells to allow better staining of the ER-resident proteins. Thus, Fig. 5d is not meant to show the membrane staining of UMOD. Please look at **Fig. 5a**, which is the live cell imaging captured by confocal microscopy without any manipulation. Fig. 5a clearly shows the membrane enrichment for WT UMOD while diffuse cytoplasmic distribution for the mutant UMOD.

---Also, brefeldin A has to be used to augment the injury, which can introduce by itself changes and non-specific noise.

BFA was used at low dose, 10 µg/mL to inhibit protein trafficking from the ER to the Golgi apparatus, thus increasing the UMOD accumulation in the ER. As shown in **Fig. 7p**, in WT cells without treatment of hrMANF, BFA itself does not induce apoptosis. In the original Fig. 5, we only applied BFA in Fig. 5F to demonstrate increased autophagy deficiency caused by more mutant UMOD accumulation in the ER with the BFA treatment. To avoid confusion, *we removed BFA-related data in the current Fig. 5*, which shows activation of ER stress, impaired autophagy/mitophagy/mitochondrial biogenesis, mtDNA leakage into the cytosol and subsequent activation of STING signaling in DEL cells without any treatment from BFA.

[Redacted] (the method is based on the paper by Malik et al. [Redacted] eLife 2018;7:e35977. DOI: <https://doi.org/10.7554/eLife.35977>).

---The position of the fused GFP (which terminus?) is not specified and could in itself produce stress.

The GFP tag is at the N terminus (we highlight it in the main text). As shown in **Fig. 5e**, only GFP-mutant UMOD, but not GFP-WT UMOD, activates the p50ATF6 branch, which mimics the finding in our mouse model. We have followed the same strategy to fuse the GFP tag at the N-terminus as what has been published by Dr. Luca Rampoldi (*PLOS ONE* 12(4): e0175970, 2017). The reason why we added an N-terminal GFP tag is for performing live cell imaging.

---The effect of recombinant MANF does not appear to be significant on p62, p-AMPK, FOXO3 (7M) and unclear if statistically significant on the readouts.

We agree with the reviewer that in previous Fig. 7M, the difference of the indicated proteins in DEL cells between absence and presence of hrMANF is not that obvious. We have tried later passages of the established

cell lines, and the difference appears to be bigger. We have replaced the previous blots in the current **Fig. 7n** with densitometry data (**Supplementary Fig. 11g**).

--- *Is the Umod release in the medium WT or mutant? shouldn't there be difference in molecular weight that reflect the transfection product?*

UMOD is released in both WT and mutant UMOD cell lines, but secretion of the UMOD is reduced in the mutant cell line. Please find the data in **Fig. 5c**. The difference in the protein migration in the cell lysates between the WT and mutant UMOD cell lines is due to underglycosylation of the mutant UMOD, most of which undergoes N-linked glycosylation in the ER, but not O-linked glycosylation in the Golgi due to ER retention. In the medium, we do not observe any difference in the molecular weight of UMOD between the WT and mutant cell lines. It appears that a small portion of the mutant UMOD can still be secreted after traveling through Golgi and thus undergoing both N- and O-linked glycosylation. The difference in the MW in the cell lysates does not derive from the 9 amino acid deletion in the mutant UMOD, which would be too small to be detected by WB.

3- MANF targeted deletion: does this lead to more Umod retention?

This is a great question. Yes, MANF deletion leads to more mutant UMOD retention in the kidneys. We have added the new data in **Fig. 6j** with quantification in **Fig. 6k**.

4- Manf overexpression:

---*what happens to Umod levels quantitatively in the kidney and urine? Manf was already upregulated in DEL/+. How much more over expression did the authors achieve in the TAL? how could they separate from a confounding effect of a protective effect from expression in other tubular segments?*

We have pooled more blots about UMOD expression in the kidneys and urines from DEL/+;MANF/P8TA and DEL/+;P8TA mice with WT littermates at age 20 wks (**Fig. 2 A-B in the letter**) in addition to the blots shown in the paper (**Fig. 7l, m**) and quantified in the **Supplementary Fig. 11e, f**, which demonstrate

66.26±4.04% reduction in the kidney mutant UMOD levels and 92.02±6.35% increase in the urinary UMOD excretion by tubular MANF overexpression in the mutant mice. In the future, we could consider checking urinary absolute UMOD levels using ELISA. But it would need to collect at least 200 µl urine for measurement in duplicates, which is quite cumbersome for us. The Wash. U mouse genetics core does mouse husbandry for us, and the core only collects no more than 50 µl urine per mouse. We have used up all previously collected urines. During the revision phase, it would be impossible for us to get urine to do urine ELISA of UMOD on the mice at age 20 wks.

As MANF is a secreted protein, it is both upregulated and excreted to the urine from MANF transgene-expressing TALs. Thus, we have performed q-PCR of [Redacted] MANF in isolated TALs from WT and DEL/+ mice without or with MANF overexpression at age 20 wks (the end of the experiment), and added the new data in the current **Fig. 7g**. If we set the average MANF mRNA level in *Umod*^{+/+};P8TA TALs as 1, the relative fold change in *Umod*^{+/+};MANF/P8TA, *Umod*^{DEL/+};P8TA and *Umod*^{DEL/+};MANF/P8TA TALs is 13.89±2.12, 4.35±2.48 and 13.9±1.59, respectively. We have noted that the total

MANF transcript level in the double transgenic mice in Fig. 7g is much higher than that in Fig. 7b. Please note that the duration of DOX administration is 16 wks (from age 4 wks-20 wks) in Fig. 7g and 4 wks in Fig. 7b, and we do not know exactly after how long of DOX treatment, MANF transcript level plateaus.

There is no TAL-specific rtTA driver mouse. Thus, what we have used is pan-tubular Pax8-rtTA mice. Recently MANF receptor neuropilin (NPTN) has been identified and published by our co-author Dr. Urano (Yagi, T et al. iScience 23, 101810, doi:10.1016/j.isci.2020.101810, 2020). The paper shows that NPTN is induced by ER stress and that MANF binds to NPTN and negates the proinflammatory effect of NPTN in pancreatic β cells. [Redacted]

---In figure 7E-H, MANF overexpression cancels activation of ATF6, whereas the suppression of BUN elevation and p62 accumulation is partial. Any thoughts about the difference? Does this mean that MANF can inhibit only a part of the signaling cascade caused by ADTKD-induced ER stress?

Yes. Based on the current Fig. 7h, it appears that MANF does not completely inhibit p50ATF6 activation in the DEL/+ kidneys.

5- The authors demonstrated that MANF overexpression is also effective for ADTKD caused by another mutation, Umod C147W/+ mice, a previously reported [Redacted] autophagy- impaired mice. However, Umod C147W/+ mice are reported to decrease LC3B protein expression, whereas Umod DEL/+ mice in this study increase LC3B expression. What does the difference mean and how do the authors reconcile their different interpretation of autophagy markers?

The reviewer is very familiar with the JCI 2017 paper published by Dr. Duffield [Redacted]

6- Figure S2, XBP1 appears to be increased?

In the **Supplementary Fig. 2b**, although 4 mutant mice show slight increase of XBP1s, the other two mutants do not. Thus, XBP1s is not increased in the mutant mice due to the big variation among different mutants. In addition, the expression of XBP1s is very weak after long exposure of the blot. We have added the densitometry data.

Minor comments:

1-Same blots of GAPDH and beta actin appear to be replicated in several figures.

Some of the blots in **Fig. 3b, c, d** share the same β -actin. For TALs, we generally cut one membrane to 2-3 pieces if the target proteins can be separated well by their MW, because isolating TALs is very labor-intensive and not much protein can be obtained from TALs per mouse. It is why the same housekeeping protein has appeared for different target proteins sometimes. But we always run the full size blot using kidney lysates if we detect a protein for the first time so that we know where the specific band is.

2- would be informative to have a supplemental data sheet with list of TAL genes differentially altered by mutation (gene name, fold change, adjusted p-value..etc)

As suggested, we have included two new **Supplementary tables** to list genes differentially altered (both upregulated and downregulated) with adjusted *p* value less than 0.05.

3- In Figure 2A, ATF6 is upregulated 12 weeks ADTKD-UMOD mice. How about 3-week-old ADTKD-UMOD mice, which accumulates the under-glycosylated form of mutant UMOD "due to ER retention" in Figure 1D?

Please find **Fig. 5 in the letter**. We do not see the difference in p50ATF6 expression between WT and the mutant mice at age 3 wks. The data suggest that accumulation of mutant UMOD may need to reach a threshold level to activate ER stress.

4- In Figure 3H, the authors showed overproduction of mitochondrial ROS in ADTKD-UMOD mice using quantitative PET/CT molecular imaging. What age of mice were used? Could decreased renal excretion affect the results if they are older than 4 weeks and impairs renal function? Have you quantified ROS in other methods?

The mice were 16 wks old and stated in the manuscript. Galuminox is a very small PET-metalloprobe, which is freely filtered by the glomeruli, regardless of the glomerular filtration rate (GFR). It is taken up by superoxide and downstream hydrogen peroxide-producing mitochondria. Thus, impaired renal function does not impact on the active accumulation of Galuminox inside mitochondria. Per the suggestion, we have also stained the kidney sections from WT and DEL/+ mice at age 16 wks with a ROS probe, chloromethyl-H₂DCFDA (Invitrogen) in **Supplementary Fig. 7b**, which is in line with the PET results.

5- In figure 3K a and b, the scale bars are missing.

Thanks for pointing it out. We have added the scale bars in the updated Fig. 3k.

6- In figure 5P, DEL shControl increases cleaved caspase as well as full-length caspase 9. Is it possible that total caspase-9 expression is increased?

The expression of cleaved caspases is generally not dependent on the level of the full-length caspases. We did not see a persistent increase of total CASP9 in DEL cells in other blots.

7- In figure 7H, does MANF treatment also correct Umod DEL/+ -induced LC3B upregulation?

Yes, we have added a new blot showing less accumulation of LC3B-II with tubular MANF overexpression in the DEL/+ kidneys in the updated Fig. 7i.

8- The molecular mechanism of autophagy of mutant Umod should be mentioned at least in the discussion part. Is the mutant UMOD ubiquitinated? Do mutant UMOD directly bind p62?

These are important questions for our future studies. We have added one paragraph as below in the discussion.

“Our study has demonstrated suppressed autophagy in the mutant TALs in ADTKD-UMOD. The molecular mechanism whereby autophagy deficiency leads to accumulation of mutant UMOD needs further investigation. The autophagy receptor p62 has an N-terminal Phox and Bem1p (PB1) domain, which mediates self-oligomerization, and a C-terminal ubiquitin-associated (UBA) domain capable of interaction with polyubiquitinated proteins. Meanwhile, p62 can bind to LC3 through its LIR (LC3-interacting region)/LRS (LC3 recognition sequence), which enables sequestration of the ubiquitinated proteins and p62 in the autophagosomes. This process is achieved efficiently by self-oligomerization of p62. We will perform further studies to elucidate whether p62 directly binds to polyubiquitinated UMOD and whether ER stress in the mutant TALs, besides its impact on p-AMPK activity, may induce upregulation of certain proteins, which can interfere with p62 binding to the ubiquitinated UMOD or LC3, thus obstructing autophagic flux.”

9- More details on the production of the mice, particularly when generating a mouse with distinct transgenic mutations of the Umod gene on separate chromosomes (Del/Cre-ET)

We have included all details of generating *Umod*^{DEL/+} mice using CRISPR under Methods. We purchased *Umod*^{CE} mice from the Jackson lab and the mice were generated by Dr. Andrew P. McMahon. We have highlighted in the manuscript that in the tamoxifen (TAM)-dependent *Umod*-driven CreER^{T2} (estrogen receptor) recombinase line (*Umod*^{IRE5 CRE-ERT2}), IRES-CRE-ER^{T2} was knocked into the 3' untranslated region (UTR) near the stop codon of the *Umod*. Thus, the CreER and the deletion mutation were knocked in on the same chromosome.

10- consider toning down the "toxic proteinopathy" in the title, since what is presented applies mostly to UMOD

In accordance with the suggestion, we have replaced “toxic proteinopathy” with “autosomal dominant tubulointerstitial kidney disease” in the title.

Reviewer #3:

This study generated a mouse model that resembles ADTKD-UMOD carrying a prevalent human mutation, and employed role of MANF in ADTKD-UMOD. The work discovered that MANF was an important regulator of autophagy/mitophagy and mitochondrial homeostasis of mutant TALs through activation of p-AMPK in ADTKD-UMOD, thereby mitigating cGAS/STING activation and promoting autophagic degradation of mutant UMOD. However, there are several concerns found in this manuscript. The concerns listed below should be addressed.

We would like to thank the Reviewer 3 for the very insightful questions and have conducted more experiments to answer these questions.

Specific comments:

1. The title of this article named “MANF stimulates autophagy and restores mitochondrial homeostasis to treat toxic proteinopathy”. However, the study only investigated one of the toxic proteinopathy, ADTKD-UMOD, which not match the title. Adding the results about other kinds of toxic proteinopathy or replacing the title would be better.

We agree with the reviewer and have replaced “toxic proteinopathy” with “autosomal dominant tubulointerstitial kidney disease” in the title.

2. The study used two kinds of animal models about MANF, including MANF deletion in mutant TALs and MANF overexpression in renal tubules mouse models. Why delete or overexpress MANF in different kidney cells? The author should discuss the reasons and provide the expression of MANF in different kidney cells under physiological and pathological status.

TAL-specific rtTA driver mice are not available. Thus, we had to use pan-tubular Pax8-rtTA mice. We have explained the reason in the revision. Recently MANF receptor neuropilin (NPTN) has been identified and published by our co-author Dr. Urano (Yagi, T et al. *iScience* 23, 101810, doi:10.1016/j.isci.2020.101810, 2020). The paper shows that NPTN is induced by ER stress and that MANF binds to NPTN and negates the proinflammatory effect of NPTN in pancreatic β cells [Redacted]

In the introduction, we have stated that “MANF is upregulated and secreted in response to ER stress” based on our and other groups’ studies. We published the first paper related to expression of MANF in kidney disease in 2016 (Kim, Y et al., Mesencephalic astrocyte-derived neurotrophic factor as a urine biomarker for endoplasmic reticulum stress-related kidney diseases. *JASN* 27: 2974–2982, 2016). In this paper, we have clearly shown that baseline level of MANF is very low in all kidney cells, and that MANF expression is induced

by ER stress. In a podocyte ER stress-induced nephrotic syndrome model, we demonstrate that MANF is specifically induced in ER-stressed podocytes. Meanwhile, in an ischemic acute kidney injury model, we prove that MANF is only upregulated in ER-stressed proximal tubular cells.

Likewise, in the current paper, we have shown that MANF is only induced in the ER-stressed mutant TALs in ADTKD based on **Fig. 6a, b, c and supplementary Fig. 9a**. In addition, per the suggestion, we have further run WB on both UMOD-negative and UMOD-positive cells from WT and DEL/+ mice at 16 wks of age to confirm that MANF is only induced in the mutant TALs. In contrast, low level of MANF expression is observed in WT TALs and UMOD-negative cells from both WT and mutant kidneys. The additional result has been added to **Supplementary Fig. 9b and shown here**.

3. Figure 4C showed highlighted kidney ingress of massive inflammatory cells and few kidney cyst. This work used PAS staining, H&E staining might reveal the inflammatory infiltration better.

Based on the reviewer's advice, we have replaced the PAS staining with H&E staining in the current Fig. 4c.

4. Result 6 showed TAL cell-specific MANF knockout caused autophagy failure and renal fibrosis. How about the mitochondrial function, renal injury and inflammation after MANF specific knockout in TAL cell?

Fig. 6i shows that kidney expression of NGAL (a renal injury marker) and fibronectin (a renal fibrosis marker) is increased after deletion of MANF from TALs. [Redacted]

[Redacted]

It is a great question regarding the role of MANF deletion on the mitochondrial function in the mutant TALs. We plan to investigate this in the near future as a stand-alone project, as the mouse breeding may take at least one year. To get the destination genotypes, we need to cross *Umod*^{DEL/+} with *Manf*^{fl/fl} first to get *Umod*^{DEL/+}; *Manf*^{fl/+} mice, which will be further crossed with *Manf*^{fl/fl} mice to get *Umod*^{DEL/+}; *Manf*^{fl/fl} mice. Meanwhile, we need to breed homozygous *Umod*^{CE/CE} mice with *Manf*^{fl/fl} to get *Umod*^{CE/+}; *Manf*^{fl/+} mice, which will be further crossed with *Manf*^{fl/fl} mice to get *Umod*^{CE/+}; *Manf*^{fl/fl} mice. After that, *Umod*^{DEL/+}; *Manf*^{fl/fl} mice will be crossed with *Umod*^{CE/+}; *Manf*^{fl/fl} mice to get desired genotypes in the littermates. As ***Manf*^{fl/fl} mice are not good breeders**, although we spent a lot of time, we were only able to get enough litters to conduct experiments in whole kidneys, **but not in TALs**. We show that p-AMPK is decreased in the mutant kidneys in the absence of MANF (**Fig. 6j**), but we were not able to observe significant difference in PGC1 α and TFAM by utilizing our frozen whole kidney lysates at 12 wks. To answer the question, we would need to breed more mice to isolate TALs, as the difference in the TALs is diluted in the whole kidney lysates, and the time spent will be well beyond the 3-6 month time window for the revision.

Although we do not have much *in vivo* data to

[Redacted]

address the question, during the last 3-4 months, [Redacted] We plan to spend more time to perform both *in vivo* (in TALs) and more *in vitro* experiments (including Seahorse and inflammation assays etc) to investigate the impact of loss of MANF in TALs on mitochondrial function, including biogenesis, OXPHOS, mitophagy, STING signaling and inflammation in a systematic manner. We hope that Reviewer 3 would understand and agree with our plan.

5. In Figure 6A, 6D and 7D, the author showed IF staining of MANF, UMOD and Hoechst 33342 on kidney paraffin sections. IF staining of MANF and TAL cell marker or tubules marker would better explain the specific knockout or overexpress MANF in TAL cells or tubules. The quantification of MANF expression in TAL cells and tubules should be helpful.

UMOD is one of the best characterized markers for TALs. Although in our *Umod* deletion mutation mice, the cellular localization of the mutant UMOD is changed, the expression of UMOD is still localized in the TALs, rather than in other tubular segments. Nevertheless, we have tried to colocalize the endogenous mouse MANF with another TAL marker, NKCC2, which is recognized by a good rabbit anti-mouse (or rat) NKCC2 Ab. However, among several anti-MANF Abs that we have purchased, we could not find a good anti-mouse MANF Ab that is not generated from rabbit. We have been using a good rabbit anti-mouse (human) MANF Ab for staining. For the same reason, we could not co-stain mouse MANF with NKCC2 in the TAL specific-MANF KO mice.

Regarding endogenous MANF expression in TALs of WT and DEL mice, we have provided additional quantitative data in **Supplementary Fig. 9b** (Please also find it under Comment 2 on Page 8). As *Manf*^{fl/fl} mice are poor breeders, due to the reason as stated above, we do not have protein lysates from TALs isolated from the WT and DEL/+ mice with or without TAL-specific MANF deletion. However, we did have one set of TALs and genomic DNA was extracted before. As shown in **Supplementary Fig. 9c**, it unambiguously proves that exon 3 of UMOD is deleted in TALs of TAL-*Manf*^{-/-} mice either on the WT or *Umod*^{DEL/+} background, which should be the most definitive method to verify that MANF is deleted in TALs. Meanwhile, WB of kidney lysates quantitatively demonstrates that MANF is depleted in the DEL/+;*Umod*^{CE/+};*Manf*^{fl/fl} kidneys (**Fig. 6e**).

For MANF transgene, which overexpresses human MANF, we are able to do co-IF staining of MANF with NKCC2 by using a goat anti-hMANF Ab and the rabbit anti-mouse NKCC2 Ab. As Pax8-rtTA mice are pan-tubular rtTA mice, we have also done additional staining of MANF with a collecting duct marker, Dolichos biflorus agglutinin (DBA) to fully characterize the overexpression pattern of transgenic MANF. The new data have been included in **Supplementary Fig. 10b**. Regarding the MANF expression levels in WT and mutant TALs with or without MANF transgene, we have added a new **Fig. 7g** by showing the mRNA levels of total MANF in TALs. MANF is a secreted protein. Thus, the overexpressed MANF in TALs is also secreted to the urine. It is the reason why we chose to do q-PCR instead of WB of MANF on TALs.

Reviewer #4:

Kim et al provided the function of MANF in the regulation of organelle homeostasis through detailed experiments. Authors demonstrated that tubular overexpression of MANF after the onset of disease stimulates autophagy/mitophagy and clearance of the misfolded UMOD, and promotes mitochondrial biogenesis through p-AMPK enhancement. In contrast, genetic ablation of endogenous MANF upregulated in the mutant mouse and human tubular cells worsens autophagy suppression and kidney fibrosis. The results of this paper are very interesting, but the author's group used the same method to generate Umod (C147W/+) KI mice, and various analyses were performed. The pathway impairment was also pointed out, and it was reported that inactivation of FOXO3 via p-AKT there is involved, and that TNF-alpha pathway is activated (J Clin Invest. 2017 Nov 1;127(11):3954-3969.). In light of this, the paper is very interesting but lacks novelty. Analysis of MANF in ADTKD patients is also needed.

We would like to thank Reviewer 4 for having thoroughly evaluated our manuscript. The novelty of this paper is that our study is the *first-of-its-kind* to discover a novel biotherapeutic protein, MANF, in the treatment of ADTKD by stimulating autophagy to eliminate the root cause of the disease (accumulation of mutant UMOD protein aggregates), and by promoting mitochondrial homeostasis through enhanced p-AMPK/FOXO3 and p-AMPK/PGC1 α activity, respectively (**US patent 11,129,871, issued in Sep. 2021**). We also demonstrate that endogenous MANF is indispensable for maintaining functional autophagy. By generating inducible MANF transgenic mice and utilizing *Manf* floxed mice, we provide the first validation *in vivo*, which will pave the way for future MANF-based therapies in ADTKD patients. Our deletion mutation mice and *Umod*^{C147W} mutant

mice share the common feature-autophagy deficiency, which further suggests that MANF may have broad therapeutic benefit in ADTKD caused by various mutations. The referred JCI paper did not study MANF at all.

There are several other major differences between our paper and the JCI paper published by Dr. Duffield, who is also a co-author in this paper and feels excited about our findings. **1.** The JCI paper did not show whether or not p-AMPK was an upstream regulator of FOXO3, and did not study whether mitophagy and mitochondrial function were compromised in the C147W mice. **2.** The JCI paper demonstrated TNF α activation, but did not determine how TNF α signaling was activated in the C147W mice. We have shown that following mitochondrial quality control failure, mtDNA leakage into the cytosol triggers STING activation, leading to production of multiple inflammatory cytokines, including TNF α , IL-1, IL-6 and ICAM1. **3.** The JCI paper provided results that TNF α receptor inhibitor can be beneficial in the C147W mice. However, other inflammatory cytokines, such as IL-1 could also be activated in the C147W mice. Thus, blocking one inflammatory cytokine may not effectively mitigate inflammation. More importantly, anti-TNF therapy would not inhibit accumulation of the mutant UMOD. We have expanded our discussion as below to highlight the advances in our study.

“It has been shown that blocking TNF α -mediated inflammation by using a soluble TNF receptor inhibitor slows disease progression in *Umod*^{C147W/+} mice⁹. Given the hierarchical order starting from mitochondrial quality control failure for the eventual activation of multiple proinflammatory cytokines in our deletion mutation mice, we envision that treatment with the upstream p-AMPK enhancer MANF, which can restore homeostasis of dysfunctional mitochondria and inhibit STING activation, would be more effective in blocking activation of a myriad of inflammatory genes, in addition to TNF α in ADTKD-*UMOD*. Furthermore, anti-TNF α treatment would not enhance autophagic clearance of the mutant UMOD protein, which is most likely the root cause of the disease.”

In **Fig. 6c**, we have shown MANF upregulation in the mutant TALs in human kidney biopsy from a patient carrying p.H177-R185 del. It is very hard to get kidney tissue from ADTKD patients, as the diagnosis of ADTKD is made by genetic testing, and medically it is not necessary to perform kidney biopsies on patients. Our co-author, Dr. Bleyer has assembled the largest cohort of ADTKD-*UMOD* patients, and fortunately is able to obtain some kidney biopsies. It is why it would be impossible for anyone to show biopsy findings from a cohort of ADTKD patients.

REVIEWERS' COMMENTS

Reviewer #2 (Remarks to the Author):

The authors responded appropriately to my comments.

Reviewer #4 (Remarks to the Author):

The authors revised well. The paper is acceptable.